# PRMT5 activates AKT via methylation to promote tumor metastasis

Lei Huang[1], Xiao-Ou Zhang[2,3], Esteban J. Rozen[1], Xiaomei Sun[1], Benjamin Sallis[4], Odette Verdejo-Torres[1], Kim Wigglesworth[1], Daniel Moon[5], Tingting Huang[1], John P. Cavaretta[1], Gang Wang[1], Lei Zhang[1], Jason M. Shohet [1], Mary M. Lee [6,7✉] & Qiong Wu [1✉]

Protein arginine methyltransferase 5 (PRMT5) is the primary methyltransferase generating symmetric-dimethyl-arginine marks on histone and non-histone proteins. PRMT5 dysregulation is implicated in multiple oncogenic processes. Here, we report that PRMT5-mediated methylation of protein kinase B (AKT) is required for its subsequent phosphorylation at Thr308 and Ser473. Moreover, pharmacologic or genetic inhibition of PRMT5 abolishes AKT1 arginine 15 methylation, thereby preventing AKT1 translocation to the plasma membrane and subsequent recruitment of its upstream activating kinases PDK1 and mTOR2. We show that PRMT5/AKT signaling controls the expression of the epithelial-mesenchymal-transition transcription factors ZEB1, SNAIL, and TWIST1. PRMT5 inhibition significantly attenuates primary tumor growth and broadly blocks metastasis in multiple organs in xenograft tumor models of high-risk neuroblastoma. Collectively, our results suggest that PRMT5 inhibition augments anti-AKT or other downstream targeted therapeutics in high-risk metastatic cancers.

[1] Department of Pediatrics, University of Massachusetts Chan Medical School, Worcester, MA, USA. [2] Program in Bioinformatics and Integrative Biology, University of Massachusetts Chan Medical School, Worcester, MA, USA. [3] School of Life Sciences and Technology, Tongji University, Shanghai, China. [4] Department of Microbiology and Physiological System, University of Massachusetts Chan Medical School, Worcester, MA, USA. [5] Department of Biology, Tufts University, Boston, MA, USA. [6] Sidney Kimmel Medical College, Thomas Jefferson University, Philadephia, PA, USA. [7] Nemours AI DuPont Children's Hospital, Nemours Pediatric Health System, Wilmington, DE, USA. ✉email: mary.lee@nemours.org; joae.wu@umassmed.edu

Metastasis has become a major barrier to the treatment of both pediatric and adult solid tumors. Contrary to tumor initiation driven by somatic mutations or genetic rearrangements, tumor metastasis typically entails epigenetic mechanisms that promote the epithelial-to-mesenchymal transitions (EMT) and cellular plasticity required for metastatic fitness[1]. A group of transcription factors (TFs), including SNAIL, ZEB, and TWIST, orchestrate the EMT program[2,3]. These EMT-TFs are induced by extracellular stimuli, including TGF-β and EGF, and by paracrine or autocrine signaling through Notch ligand and Wnt, as well as by the hypoxic tumor microenvironment and interactions with tumor-associated stroma[4–6]. On the molecular level, the EMT process is modulated by multiple signaling pathways, some of which are mediated by phosphoinositide 3-kinase (PI3K)/protein kinase B (AKT)/mammalian target of rapamycin (mTOR)[7,8].

Epigenetic regulators such as the polycomb repressive complexes (PRCs), histone acetyltransferases (HATs) and deacetylases (HDACs), histone methyltransferases (HMTs) and demethylases (HDMs), DNA methyltransferases (DNMTs), nucleosome remodelers and in particular, protein arginine methyltransferase 5 (PRMT5), have been shown to participate by regulating EMT-TFs and EMT markers[9]. For example, the PRMT5-MEP50 complex methylates H3R2 to induce EMT-TFs activator genes; and methylates H4R3 to repress metastasis suppressor genes E-cadherin and GAS1 in lung and breast cancer[10].

Neuroblastoma is a neural crest-derived embryonal malignancy in infants that accounts for almost 15% of all pediatric cancer deaths[11]. This highly aggressive cancer arises within the paraspinal sympathetic ganglia and adrenal glands and is thought to derive from dedifferentiated sets of a primitive neural crest that fails to differentiate due to epigenetic alterations[11–15]. Recent single-cell analyses have demonstrated remarkable intra-tumoral plasticity and shown neuroblastoma to contain both mesenchymal and more differentiated adrenergic cell population[16,17]. It has also been demonstrated that epigenetically defined transcriptional networks distinguish highly tumorigenic mesenchymal subpopulations[18]. During neural crest development, AKT signaling regulates EMT by multiple mechanisms and has been implicated in cancer-specific EMT, invasion, and metastasis[19–22]. Based on these observations, approaches to target EMT in metastatic tumors such as neuroblastoma should prove clinically valuable.

We report now that PRMT5-mediated AKT1 arginine 15 methylation is required for its activation. PRMT5 promotes AKT1 interaction with phosphatidylserine, translocation to the plasma membrane, and interaction with upstream kinases PKD1 and mTORC2. Furthermore, PRMT5/AKT signaling controls the expression of essential EMT transcription factors controlling tumor cell migration and invasion. We show that a PRMT5 inhibitor markedly reduces primary tumor growth and attenuates tumor cell metastasis to the liver, a common metastatic site for neuroblastomas. In a metastasis mouse model, PRMT5 depletion blocks liver and lung metastasis. Collectively, our results indicate that PRMT5 inhibition may represent a therapeutic tool to prevent metastatic cancers.

## Results

### PRMT5 is dysregulated in advanced-stage neuroblastoma.
To explore the clinical significance of PRMT5, we analyzed its expression in the RNA-seq results from 461 patients with stage 1, stage 2, stage 3, and stage 4 neuroblastoma[23]. PRMT5 transcript levels were significantly higher in stage 3 and stage 4 patients than in stage 1 patients (Fig. 1a). We also performed this analysis in three additional independent neuroblastoma annotated patient cohorts available in the R2 database (R2: Genomics analysis and visualization platform [http://r2.amc.nl]). We found that PRMT5 transcripts were more abundant in stage 4 than stage 3 patients and negatively correlated with patient survival (Fig. 1b–d).

### PRMT5 inhibition reduces neuroblastoma cell viability.
Given its involvement in many cancers, PRMT5 has become a promising target for therapeutic intervention and PRMT5 inhibitors are currently being tested in clinical trials for advanced solid tumors as well as myeloid leukemia and non-Hodgkin's lymphoma[24,25]. For in vivo studies, we used GSK3326595 (GSK595, hereafter) due to its confirmed efficacy in a breast cancer mouse model[26]; for in vitro studies, we used its sister compound GSK3302591 (GSK591, hereafter) that shows increased cell permeability in tissue culture. PRMT5 inhibition by GSK591 in the low nanomolar range significantly decreased the viability of three neuroblastoma cell lines representing the clinical diversity of neuroblastoma (Fig. 1e). PRMT5 generates the majority of cellular symmetric dimethylarginine (SDMA); thus, we examined global SDMA as a readout of PRMT5 enzymatic activity. We verified that GSK591 treatment significantly decreased SDMA in a dose-dependent manner, indicating the on-target effects of this compound (Supplementary Fig. 1a). We generated doxycycline-inducible shRNA to knock down PRMT5 expression in neuroblastoma cell lines (Supplementary Fig. 1b). Induced depletion of PRMT5 decreased global SDMA (Supplementary Fig. 1c) and this result was further confirmed with an independent SDMA antibody (Supplementary Fig. 1d). In addition, PRMT5 knockdown significantly impaired neuroblastoma cell viability (Fig. 1f). GSK591 treatment triggered apoptosis in a dose-dependent manner, as evidenced by the increasing levels of cleaved caspase-3 (Fig. 1g), explaining the observed reduction in viability. More condensed, pycnotic nuclei were seen in GSK591-treated cells than in control cells, as visualized by Hoechst 33342 (Fig. 1h). When stained with a green fluorescent dye recognizing cleaved caspase-3/7, GSK591-treated cells had a higher percentage of caspase-3/7 positive cells (Fig. 1i, j and Supplementary Fig. 1e).

### PRMT5 inhibition attenuates primary tumor growth and blocks hepatic metastasis.
We then tested the efficacy of GSK595 on primary tumor growth in an orthotopic mouse xenograft model. To monitor the tumor growth by live animal imaging, we transduced CHLA20 or NGP neuroblastoma cells with iRFP720-Luc reporter lentiviral vectors that allowed in vivo imaging by bioluminescence and differentiated human tumor cells from mouse cells by iRFP720. We implanted neuroblastoma cells into the renal capsule of NOD/SCID mice. This is a well-characterized model that faithfully recapitulates the aggressive, highly vascular neuroblastoma growth from adrenal parasympathetic precursors, as well as the local invasiveness and metastatic spread observed in human neuroblastoma[27]. Animals were treated for two weeks with 100 mg/kg GSK595 or with a vehicle by oral gavage twice daily (Fig. 2a). Serum blood chemistries showed no significant toxicity (Supplementary Table 2), and there was no severe weight loss after two weeks of GSK595 treatment (Supplementary Fig. 2a).

We found that the tumor mass from the GSK595-treated group was significantly reduced in xenografts of CHLA20 (Fig. 2b) and NGP (Fig. 2e) compared to control mice. The effective delivery of GSK595 to xenograft tumors was confirmed by examining SDMA levels in tumor tissues from control and GSK595-treated mice (Supplementary Fig. 2b, c). Light microscopy showed that the size of the ex vivo tumors from the vehicle-treated mice was much larger than that from GSK595-treated mice in both CHLA20 and NGP xenografts, respectively (Fig. 2c, f). Bioluminescent imaging demonstrated that there were fewer tumor cells in the GSK595-treated tumor tissues compared to vehicle-treated mice

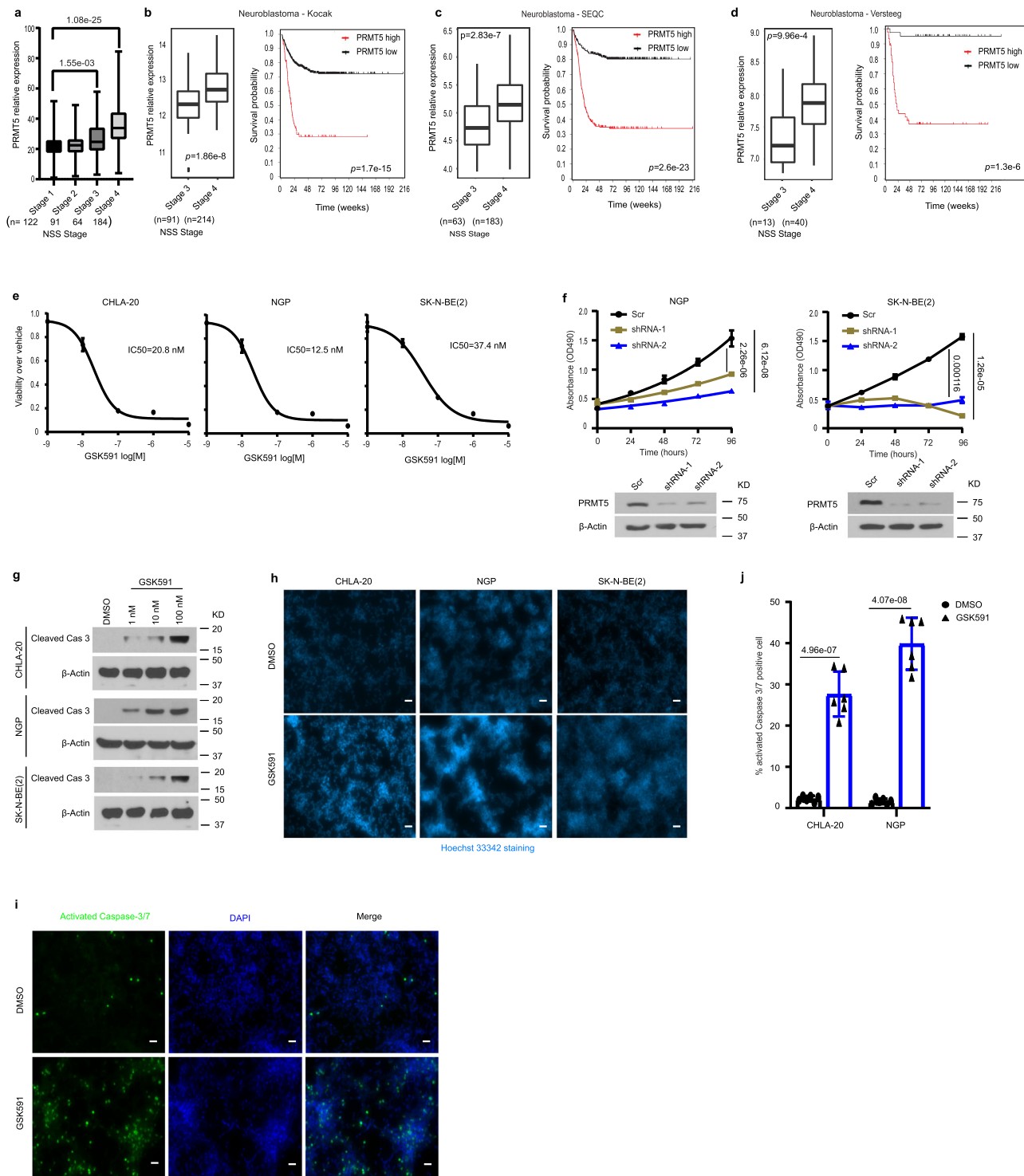

(Fig. 2d, g). Importantly, we also found fewer tumor cells in the livers of mice treated with GSK595 than in control mice by bioluminescent imaging (Fig. 2h). This observation was confirmed by FACS analysis of iRFP720 positive human neuroblastoma cells that were implanted (Fig. 2i, j). Collectively, GSK595 treatment suppresses primary tumor growth and blocks tumor cell metastasis to the liver.

**PRMT5 inhibition impairs AKT signaling**. PRMT5 post-translationally modifies proteins, hence modulating their

biological functions[24,28–30]. We explored the consequences of PRMT5 inhibition on global signaling networks in neuroblastoma cells treated with DMSO or GSK591 by a proteomics-based targeted screen, an assay that is a similar approach to proximity ligation assay in a high throughput manner to measure the expression, modification, and cleavage of crucial proteins involved in 20 signal transduction pathways (https://www.activsignal.com/). These data confirmed several results shown above (Fig. 1g) and revealed additional changes including increased phospho-H2AX Ser139 and cleaved Parp, hallmarks of DNA damage and apoptosis, respectively. In addition, we

**Fig. 1 Overexpression of PRMT5 is associated with high-risk neuroblastoma and poor patient survival and is required for neuroblastoma cell proliferation in vitro. a** The RNA expression of PRMT5 in increasing grades of neuroblastoma from the GSE49711 RNA-seq data series (Stage 1, $n = 122$, Stage 2, $n = 91$, Stage 3, $n = 64$, Stage 4, $n = 184$). Kruskal–Wallis with Dunn's multiple comparisons test was used to determine $p$ values, which were corrected with the Benjamini–Hochberg method.). **b–d** PRMT5 expression levels (left) and patient survival probability (right) in stage 3 and stage 4 neuroblastoma patients in Kocak (**b**, Stage 3, $n = 91$, Stage 4, $n = 214$), SEQC (**c**, Stage 3, $n = 63$, Stage 4, $n = 183$), and Versteeg (**d**, Stage 3, $n = 13$, Stage 4, $n = 40$) databases. $P$ values were calculated with a two-sided Wilcoxon rank-sum test for boxplots (left) and $p$ values were calculated with a log-rank test for survival curves (right). **a–d** Boxplot center represents mean, the box represents SD, and whiskers represent minimum and maximum. **e** The efficacy of PRMT5 small molecule inhibitor GSK3203591 (GSK591) was determined by MTS assay in CHLA20, NGP, and SK-N-BE (2) cells ($n = 3$). The IC50 value was determined by nonlinear regression (curve fit) using $\log_{10}$ (inhibitor) versus response (three parameters) model in GraphPad Prism. **f** Cell viability was measured by MTS assay in a scramble or shRNA targeting PRMT5 in NGP (left) and SK-N-BE (2) cells (right) ($n = 4$). **g** Cleaved caspase-3 levels in neuroblastoma cell lines treated with increasing doses of GSK591. **h** Hoechst 33342 staining in cells treated with DMSO or 100 nM GSK591. Scale bars, 100 μm. **i** Apoptosis was measured by caspase-3/7 staining in CHLA20 cells treated with DMSO or GSK591. **j** Quantification of caspase-3/7 positive cells ($n = 6$). **f**, **j** $p$ values were indicated by a two-tailed unpaired Student's $t$-test using Microsoft Excel; error bars represent SD (**e–j**). **e–i** Representative results from three independent experiments. Uncropped immunoblots are provided in the Source Data file.

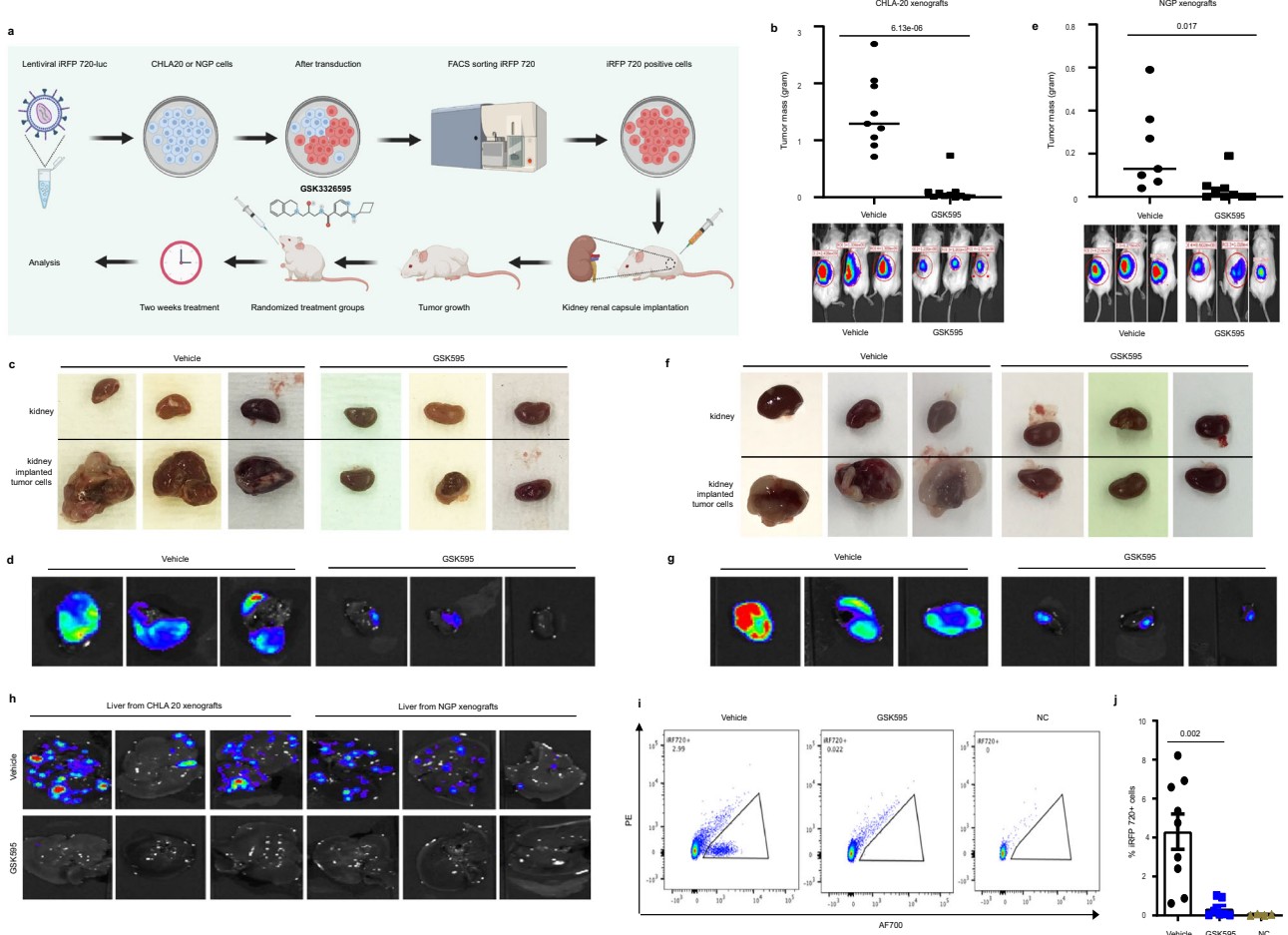

**Fig. 2 Inhibiting PRMT5 methyltransferase activity attenuates primary tumor growth and metastasis. a** Schematic diagram of PRMT5 inhibitor GSK3326595 (GSK595) in vivo study in the kidney renal capsule implantation xenograft model. **b** Tumor mass of CHLA20 iRFP720-Luc xenograft tumors from mice treated with vehicle ($n = 9$) or GSK595 ($n = 10$) (top) and representative in vivo bioluminescent images and quantification of tumor (bottom). Tumor mass was calculated by subtracting the weight of the normal kidney from the weight of the kidney that was implanted with tumor cells. Representative ex vivo images of the tumor by light microscope (**c**) or bioluminescence (**d**). **e** Tumor mass of NGP iRFP720-Luc xenograft tumors from mice treated with vehicle ($n = 7$) or GSK595 ($n = 9$) (top) and representative in vivo bioluminescent images and quantification of tumor (bottom). Representative ex vivo images of the tumor by light microscope (**f**) or bioluminescence (**g**). **h** Representative ex vivo bioluminescent images of liver from mice bearing CHLA 20 or NGP xenograft tumors treated with vehicle or GSK595. **i** Representative charts of FACS analysis of iRFP720 positive human neuroblastoma cells in hepatocytes isolated from the whole liver from CHLA20 xenografted mice, NC negative control. **j** Quantification of the percentage of iRFP720 positive tumor cells determined by flow cytometry in hepatocytes isolated from the whole liver from CHLA20 xenografted mice. Vehicle, $n = 9$, GSK595, $n = 7$, NC, $n = 4$. $p$ values were indicated by a two-tailed unpaired Student's $t$-test using Microsoft Excel; error bars represent SD (**b**, **e**, **j**).

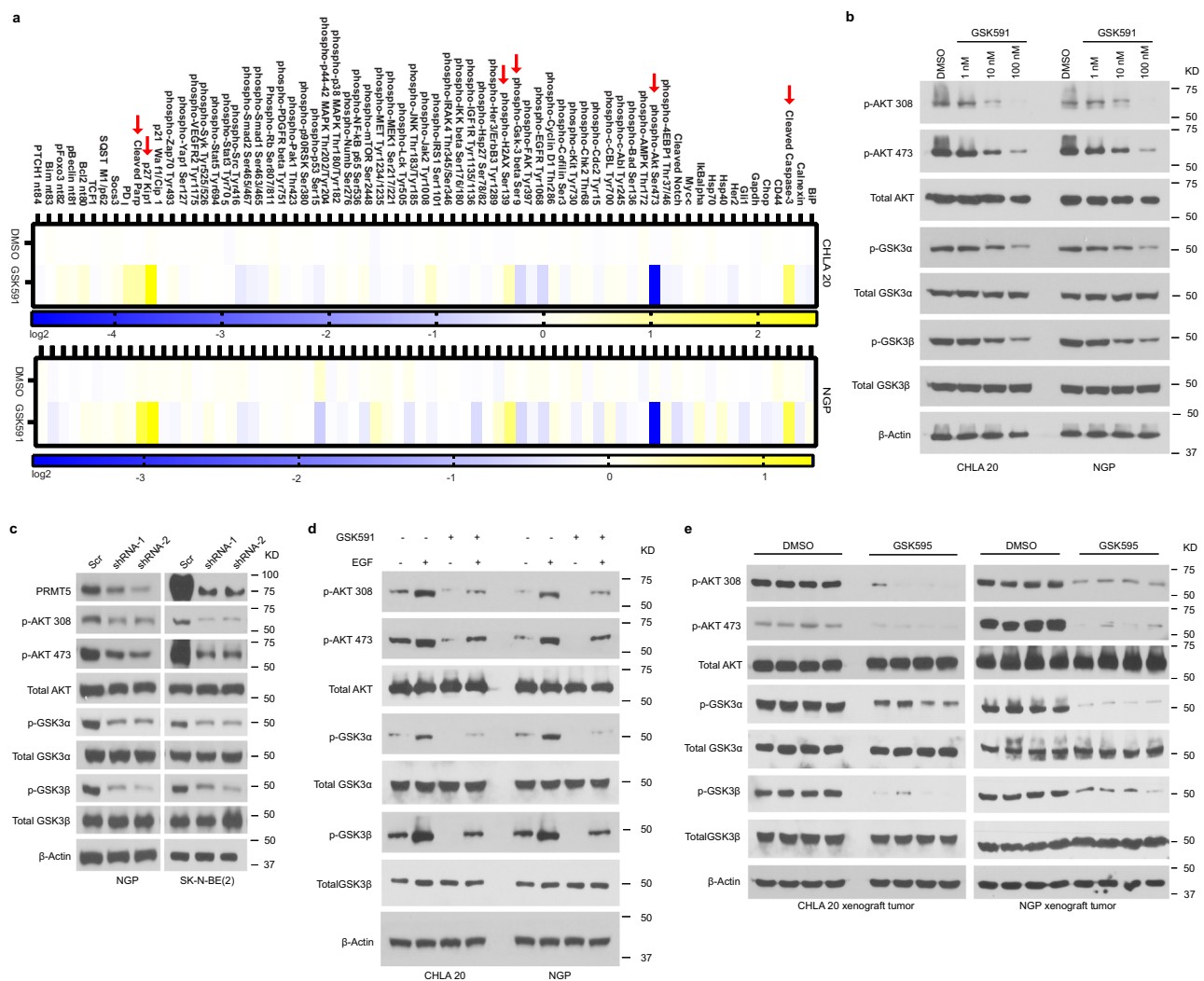

**Fig. 3 PRMT5 inhibition impairs AKT signaling. a** Proteomics-based pathway screening by IPAD platform (immuno-paired-antibody detection assay) for the expression or modification of key proteins involved in more than 20 signaling pathways. Signals were normalized to internal levels of GAPDH and beta Tubulin. Heatmap showed differences between DMSO and GSK591 groups based on the average value from two independent experiments ($n = 2$ biologically independent samples). The color scale represented log2 fold changes over DMSO treatment. **b** Immunoblots showing phosphorylation of AKT, and its downstream targets phospho-GSK3α and phospho-GSK3β in CHLA20 and NGP cells treated with DMSO or increasing doses of GSK591. **c** Western blots of AKT phosphorylation and AKT downstream targets phospho-GSK3α and phospho-GSK3β in the presence and absence of PRMT5 in NGP or BE2 cells harboring scramble or shPRMT5. **d** Levels of phosphorylated AKT, GSK3α, and GSK3β with or without EGF stimulation in DMSO and GSK591-treated CHLA20 and NGP cells. **e** Activated AKT and its targets phospho-GSK3α and phospho-GSK3β were detected by Western blotting in xenograft tumors from vehicle or GSK595-treated mice. **b–d** Representative results from three independent experiments. Uncropped immunoblots are provided in the Source Data file.

observed dramatically increased levels of the cell cycle inhibitor p27. Most strikingly, the single most affected target was phospho-AKT Ser473 in both CHLA20 and NGP cell lines (Fig. 3a). Upon treatment with GSK591, both phospho-AKT Thr308 and Ser473 were significantly diminished in a dose-dependent manner (Fig. 3b). In addition, phosphorylation of GSK3α and GSK3β, known downstream targets of AKT, was markedly decreased (Fig. 3b). These findings were reproduced upon PRMT5 knockdown via shRNA (Fig. 3c).

Next, we examined AKT activation in cells treated with DMSO or PRMT5 inhibitor GSK591 with or without EGF stimulation. As shown in Fig. 3d, both basal and stimulated AKT phosphorylation was suppressed in the presence of GSK591. Moreover, phospho-AKT Thr308 and phospho-AKT Ser473 staining was diminished in GSK591-treated cells examined by immunofluorescence, while total-AKT staining remained

unchanged, serving as a matched control for this experiment (Supplementary Fig. 3a). Importantly, AKT phosphorylation was also attenuated in tumors from GSK595-treated mice in both xenograft models (Fig. 3e).

**PRMT5 inhibition diminishes SDMA on AKT.** Aberrant activation of AKT is driven by oncogenic activation of RAS and PI3K or by loss of PTEN in many types of cancer[31]. The activity of AKT depends on the dynamic balance between phosphorylation by kinases, such as PDK1, mTORC2, DNA-PK, and dephosphorylation by phosphatases, such as PP2A and PHLPP1/2[32–37]. PI3K and PTEN also modulate the amount of phosphatidylinositol (3,4,5)-trisphosphate (PIP$_3$), further affecting AKT activation[21,22]. We, therefore, hypothesized that PRMT5 inhibition might indirectly regulate AKT activation via these known upstream regulators. However, we failed to detect any changes in

these proteins upon GSK591 treatment in neuroblastoma cells (Supplementary Fig. 3b, c). Even under EGF stimulation, we did not detect any differences in levels of PDK1, phosphorylated PDK1, Rictor, or PTEN (Supplementary Fig. 3d). It has been well documented that growth factor-mediated receptor signaling leads to AKT activation. We, therefore, tested the expression and phosphorylation of a group of well-known receptor tyrosine kinases (RTKs), including EGFR, IGF1R, VEGFR, and ERBB3. Of these, only EGFR was affected by GSK591 treatment (Supplementary Fig. 3e). However, overexpressing EGFR in GSK591-treated cells did not rescue AKT phosphorylation (Supplementary Fig. 3f). Furthermore, the combination of EGFR inhibitor Erlotinib with GSK591 failed to show any additive effect on AKT phosphorylation (Supplementary Fig. 3g). Taken together, these results suggest that the decrease of AKT phosphorylation may not be caused by indirect downregulation of EGFR expression under PRMT5 inhibition.

Next, we considered the possibility that PRMT5 could directly methylate AKT. AKT has three isoforms (AKT1, AKT2, and AKT3) encoded by different genes that share a conserved N-terminal domain, a central kinase domain, and a C-terminal regulatory domain[38]. AKT1 is ubiquitously expressed, AKT2 is primarily expressed in insulin-responsive tissues, while AKT3 is highly expressed in the brain and testes[38]. We examined the abundance of AKT isoforms in neuroblastoma cells and found they were all highly expressed (Supplementary Fig. 4a). Interestingly, we detected an interaction between PRMT5 and AKT by co-immunoprecipitation (co-IP) (Fig. 4a). In GSK591-treated neuroblastoma cells, SDMA, as well as phosphorylation on endogenous AKT1 was markedly decreased (Fig. 4b). In contrast, there was no significant change of SDMA and phosphorylation of AKT2 and AKT3 in the presence of GSK591 when pulled down with isoform-specific antibodies (Supplementary Fig. 4b, c). Importantly, the re-introduction of wild-type PRMT5 but not catalytically inactive mutant PRMT5 to PRMT5 knockdown cells or GSK591-treated cells completely rescued the decreased AKT phosphorylation (Fig. 4c, d). These results suggest a direct role of PRMT5 in regulating AKT phosphorylation via methylation.

**PRMT5-mediated AKT1 arginine 15 methylation is required for AKT1 phosphorylation**. Liu et al. reported that PRMT5-induced methylation prevented GSK3β-mediated phosphorylation of SREBP1a on S430, leading to its degradation through the ubiquitin-proteasome pathway[30]. In our study, PRMT5 inhibition did not affect the total-AKT protein levels (Fig. 3b–d), thereby excluding the possibility that protein degradation reduced AKT phosphorylation. Hence, the decrease of phospho-AKT might be a direct consequence of hypo-methylation under PRMT5 inhibition that impairs subsequent phosphorylation. PRMT5 preferentially methylates arginine residues within arginine- and glycine-rich (RGG/RG) motifs[29]. Scanning the AKT1 sequence, we identified Arg 15 as a potential methylation site and mutated this arginine to lysine by site-directed mutagenesis (R15K). To verify this potential PRMT5 methylation site, we performed an in vitro methylation assay, where HA-tagged AKT1 wild type or R15K mutant were purified from CHLA20 cells and incubated with $^{14}$C labeled methyl donor S-adenosyl methionine (SAM) with or without recombinant PRMT5/MEP50. We detected a $^{14}$C signal in AKT1 wild-type incubated with $^{14}$C-SAM in the presence but not in the absence of PRMT5/MEP50, indicating the $^{14}$C-SAM was transferred by the addition of exogenous PRMT5/MEP50 (Fig. 4e). This result supports that PRMT5 methylates R15 of AKT1.

Next, we asked whether PRMT5-mediated AKT1-R15 methylation impacts its activation. HA-tagged AKT1 wild-type or R15K

mutant constructs were transfected into neuroblastoma cells. After immunoprecipitation with anti-HA antibody-conjugated beads, SDMA on HA-tagged proteins was examined by Western blotting. We found that SDMA on AKT1-R15K mutant was nearly abolished, indicating Arg 15 is the predominant methylation site of PRMT5 (Fig. 4f, left). When stimulated with EGF, the SDMA on AKT1-R15K was reduced compared to AKT1 wild type (Fig. 4f, right). Notably, both basal and stimulated phosphorylation of Thr308 and Ser473 on AKT1-R15K were suppressed (Fig. 4f). AKT activation involves two sequential phosphorylation events, including priming by phosphorylation of Thr308 and full activation by phosphorylation of Ser 473[32,37]. In the AKT1-R15K mutant, both phosphorylation events were severely compromised. Our results suggest that PRMT5-mediated AKT1-R15 methylation is required for AKT1 activation.

**AKT1-R15 methylation promotes AKT1 association with plasma membrane and interaction with upstream kinases**. The activation of AKT relies on its Pleckstrin-homology (PH) domain binding to phospholipids in the plasma membrane, leading to conformational changes to enable its activation by phosphoryation[39]. To better understand the impact of AKT1-R15 methylation on its association with the plasma membrane, we examined the levels of activated AKT1 in the membrane-bound fraction with or without EGF stimulation in GSK591-treated cells as well as PRMT5 knockdown cells. We found that basal and stimulated phospho-AKT1 phosphorylation was markedly reduced in GSK591-treated cells (Fig. 5a), nearly undetectable in the membrane fraction in PRMT5 knockdown cells (Fig. 5b). Moreover, the colocalization of activated AKT1 wild type with the plasma membrane was detected under basal and EGF stimulated conditions (Fig. 5c). In contrast, the AKT1-R15K mutant failed to localize to the plasma membrane, even in the presence of EGF stimulation (Fig. 5c). Huang BX, et al. reported that AKT1-R15 and -K30 are essential for its association with phosphatidylserine (PS) to promote its phosphorylation[40]. We performed in vitro lipid binding assay where purified HA-AKT1 wild type or R15K mutant was incubated with PS-bound agarose beads. After incubation, PS beads were washed and proteins bound to the beads were eluted and analyzed by Western blotting. As shown in Fig. 5d, AKT1 wild type was detected in the PS beads-bound fraction, whereas AKT1-R15K mutant mostly remained in the flowthrough fraction. These results reveal that AKT1-Arg 15 methylation is essential for its translocation to the plasma membrane.

PDK1 and mTORC2 catalytic subunit SIN1 are well-defined upstream kinases that phosphorylate AKT threonine 308 and serine 473[32,37]. We asked whether PRMT5-mediated AKT1-Arg 15 methylation is required for the interaction with these kinases. HA-tagged AKT-wild type or AKT1-R15K were immunoprecipitated from transfected neuroblastoma cells and analyzed by western blotting for PDK1 and SIN1. As shown in Fig. 5e, HA-AKT1-R15K failed to bind PDK1 and SIN1, even under EGF stimulation. Furthermore, we examined whether the interaction of endogenous AKT1 with PDK1 and SIN1 was impaired upon PRMT5 inhibition. As shown in Fig. 5f, g, the association of endogenous AKT1 with PDK1 and SIN1 diminished in GSK591-treated cells as well as PRMT5 knockdown cells. In summary, these data suggest a pivotal role of PRMT5-mediated AKT1-Arg 15 methylation in AKT1 relocation to the plasma membrane and interaction with PDK1 and mTORC2 for its subsequent activation.

**The PRMT5/AKT axis regulates EMT pathways**. Previous studies have shown that AKT acts downstream of TNFα and TGFβ

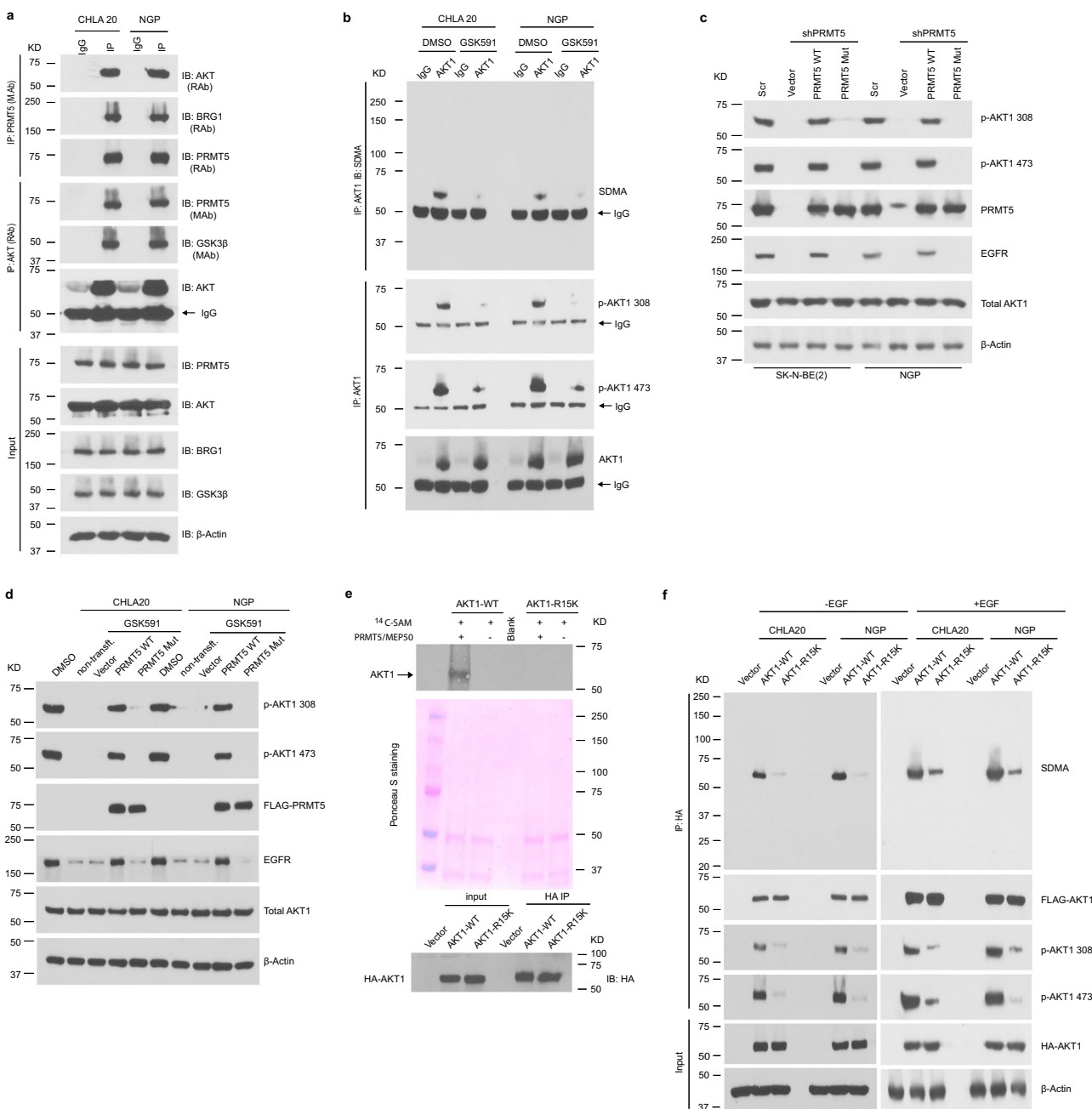

**Fig. 4 PRMT5 methylates AKT. a** PRMT5/AKT interaction captured by co-immunoprecipitation (co-IP). The lysate was immunoprecipitated with anti-PRMT5 antibody followed by immunoblotting with anti-AKT antibody in CHLA20 and NGP cells (top), and the reciprocal co-immunoprecipitation was shown in the middle. BRG1 and bait protein PRMT5 were shown as positive controls for the PRMT5 IP, whereas GSK3β and bait protein AKT served as positive controls for AKT IP. The input was used as internal controls (bottom). **b** Immunoprecipitation of AKT1 followed by a Western blotting analysis of symmetric dimethylarginine (SDMA) of AKT1, phosphorylation of AKT1 on Thr308 and Ser473 in DMSO or GSK591-treated CHLA20 and NGP cells. **c** AKT1 phosphorylation was detected by immunoblotting in a scramble or PRMT5 knockdown cells when forced expressing a wild-type PRMT5 or an enzymatic deficient form of PRMT5. **d** Analysis of AKT1 phosphorylation in DMSO or GSK591-treated CHLA20 and NGP cells expressing PRMT5 wild type or enzyme dead mutant. **e** In vitro methylation assay showing the methylation of AKT1 wild type and R15K mutant by recombinant PRMT5/MEP50 (top), Ponceau S staining of the membrane showing equal loading of each sample (middle), and Western blotting analysis showing an equal amount of HA-tagged proteins pulled down by anti-HA beads (bottom). **f** Analysis of SDMA and phosphorylation of AKT1 wild type or R15K mutant in CHLA20 and NGP cells with (right) or without (left) EGF stimulation. Cells were transfected with AKT1 wild type or R15K mutant. In the case of EGF stimulation, 24 h post-transfection, cells were serum-starved overnight and then treated with 10 ng/mL EGF for 15 min before harvest. **a–d, f** Representative results from three independent experiments. **e** Representative results from two independent biological samples. Uncropped immunoblots are provided in the Source Data file.

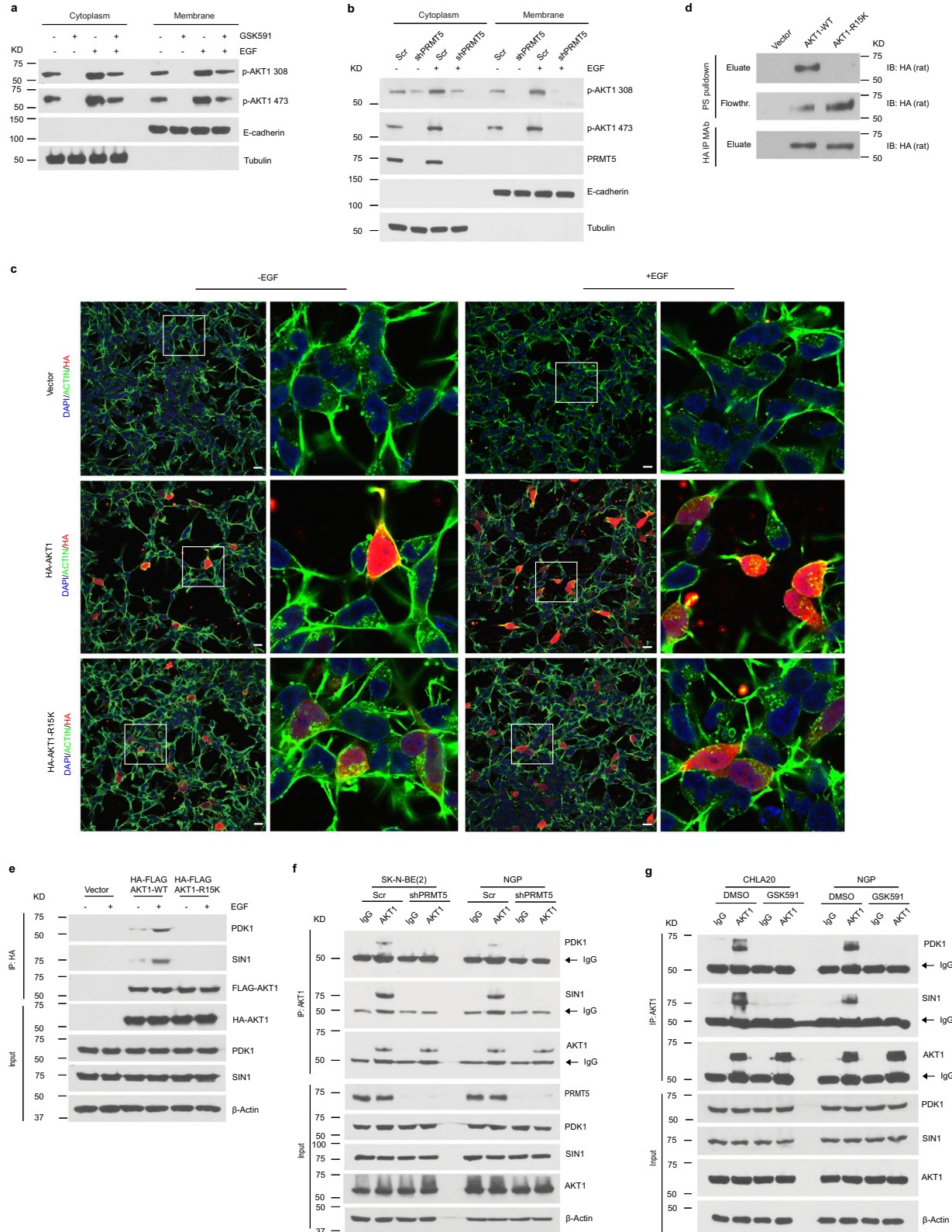

to regulate the transcriptional activity of EMT transcription factors (EMT-TFs) such as SNAIL and TWIST1[41–43]. Given that AKT activation is compromised upon PRMT5 inhibition, we speculated that the EMT program is likely attenuated. Indeed, the expression of ZEB, TWIST1, and SNAIL decreased in cells treated with GSK591 in a dose-dependent manner (Fig. 6a). Moreover, protein levels of these EMT-TFs were reduced when PRMT5 was

knocked down (Fig. 6b). Importantly, diminished levels of ZEB, TWIST1, and SNAIL were detected in neuroblastoma xenografts from mice treated with GSK595 compared to controls (Fig. 6c).

We next performed assays to explore the impact of the decreased EMT on cell migration and invasion. We used would healing to assess cell migration after eliminating the confounding effect of cell proliferation, and observed that cell migration was

**Fig. 5 PRMT5-mediated AKT1-R15 methylation is required for its activation. a** Western blotting analysis of the distribution of activated AKT1 in cytosolic or membrane fractions from DMSO or GSK591-treated CHLA20 cells in the absence or presence of EGF. **b** The presence of activated AKT1 in cytosolic and membrane fractions from scramble or PRMT5 knockdown cells with or without EGF stimulation. **c** Colocalization of AKT1 wild type or R15K mutant with plasma membrane by immunofluorescence visualized under confocal microscopy in CHLA 20 cells in the absence or presence of EGF. Scale bars, 100 μm. **d** The binding to phosphatidylserine (PS) of AKT1 wild type or R15K mutant was examined by the presence of HA-tagged protein in the eluate or flowthrough fraction after incubation with phosphatidylserine coated agarose beads by Western blotting. Top, elution from PS beads; middle, flowthrough after incubation; bottom, HA-tagged protein eluted from anti-HA beads. **e** The association of AKT1 wild type or R15K mutant with PDK1 or mTORC2 was analyzed by Western blotting in CHLA20 with or without EGF stimulation. Exogenous AKT1 wild type or R15K mutant was pulldown by HA beads, and the precipitants were analyzed by Western blotting for PDK1 and SIN1. **f** The interaction of AKT1 with PDK1 and mTORC2 was measured by immunoprecipitation of endogenous AKT1 followed by Western blotting against PDK1 and SIN1 in SK-N-BE(2) and NGP cells with or without PRMT5 knockdown. **g** The recruitment of PDK1 and mTORC2 was examined in DMSO or GSK591-treated CHLA20 and NGP cells. **a–g** Representative results from three independent experiments. Uncropped immunoblots are provided in the Source Data file.

attenuated in GSK591-treated cells compared to control cells (Fig. 6d, e). Most importantly, cell invasion through the extracellular matrix was significantly reduced under GSK591 treatment in an in vitro transwell invasion assay (Fig. 6f, g). These results are consistent with our in vivo studies, where PRMT5 inhibition potently blocked metastasis to the liver. If PRMT5 regulates EMT via AKT1 activity, restoring AKT1 activation should rescue the decreased expression of EMT-TFs. We transfected GSK591-treated cells with either vector or constitutively activated AKT1 (myr-AKT1) and analyzed the expression of EMT-TFs. As shown in Fig. 6h, myr-AKT1 overexpression was sufficient to restore the protein levels of TWIST and SNAIL, suggesting the downregulation of these proteins was a direct consequence of impaired AKT1 activation. Moreover, overexpressing PRMT5 wild type but not the enzymatic deficient form of PRMT5 increased the protein levels of TWIST1 and SNAIL and concomitantly promoted cell invasion (Fig. 6i, j). Collectively, our results suggest that PRMT5/AKT1 axis promotes metastasis by enhancing the expression of EMT-TFs.

**PRMT5 depletion blocks tumor cells metastasis.** To rule out the possibility that reduced metastasis was conveyed by insufficient primary tumor growth, we tested SK-N-BE(2) neuroblastoma cells expressing iRFP720 and doxycycline-inducible PRMT5 shRNA in a metastatic mouse model. After tail vein injection into NOD/SCID mice, we compared metastasis in control animals relative to animals treated with doxycycline to induce PRMT5 knockdown. Metastasis was measured using in vivo bioluminescent imaging, ex vivo bioluminescent and fluorescent imaging (iRFP720), and flow cytometry analysis of iRFP720 expression in cells isolated from multiple organs (liver, lung, kidney, and bone marrow). As shown in Fig. 7a, five out of six control mice had multiple metastases detected by live animal imaging of bioluminescence, while minimal to no metastases were detected in the doxycycline-treated PRMT5 knockdown cohort.

Neuroblastoma commonly metastasizes to the bone marrow, liver, and other organs[44]. Ex vivo organ evaluation showed metastatic neuroblastoma cells in the livers and lungs of all the control mice, and no or minimal luciferase or fluorescent signals in the doxycycline-treated cohort (Fig. 7b, c). We also observed weak but significant bioluminescence and iRFP720 expression in the kidneys of control mice (Supplementary Fig. 5a). Two control mice showed also a visible signal in the femur (Supplementary Fig. 5c). When quantified by flow cytometry, 4% of the liver cells isolated from control mice were iRFP720 positive, whereas no significant iRFP720 positive tumor cells were detected from mice on doxycycline treatment (Fig. 7d). Similar results were observed in the lung, where 4.5% of lung cells were iRFP720 positive isolated from control mice (Fig. 7e). We also detected significantly higher iRFP720 positive kidney cells from the control group (Supplementary Fig. 5b). There was a trend but

not a statistically significant difference in the bone marrow cells when analyzed for the presence of iRFP720 cells (Supplementary Fig. 5d). Taken together, these data indicate a significant inhibition of neuroblastoma metastatic spread upon PRMT5 knockdown that is independent of primary tumor growth. These data were consistent with our findings that GSK595 treatment of primary xenografts blocked liver metastasis (Fig. 2).

**Discussion**

As noted, there is an increasing interest in the pharmacological targeting of PRMT5, as it is highly over-expressed in multiple aggressive metastatic cancers[24,25,45]. We illustrate that PRMT5 inhibition not only attenuates primary tumor growth, but also potently blocks metastasis in both orthotopic implantation and metastatic models (Figs. 2, 7). We show that PRMT5 modulates AKT1 activation via direct methylation of Arg 15 (Figs. 3–5) and that AKT signaling regulates the expression of essential EMT-TFs that orchestrate the EMT program responsible for tumor metastasis (Fig. 6). In line with the inhibition of liver metastasis in xenograft models, we confirm that PRMT5 inhibition decreases the expression of EMT-TFs and impairs cell migration and invasion (Fig. 6). Furthermore, PRMT5 depletion blocks tumor metastasis in the liver, lung, and kidney, independent of primary tumor growth (Fig. 7). These findings highlight the critical role of PRMT5 in the process of metastasis. Specifically, we show that PRMT5-mediated Arg 15 methylation of AKT1 is required for AKT1 activation (Fig. 4). We show that loss of symmetric dimethylation of Arg 15 within the amino-terminal PH domain prevents AKT1 binding to phosphatidylserine and relocation to the plasma membrane and blocks the interaction with PDK1 and mTORC2. As a result, AKT1 phosphorylation on Thr308 and Ser473 is diminished. Thus, this posttranslational modification is crucial for AKT1 activation and its biological function (Fig. 8).

Discovered almost three decades ago, many downstream effectors of AKT have been characterized, such as GSK3, mTOR, and FOXO[21,22]. In contrast, only a few upstream regulators have been confirmed, including PIP3 modulators PI3K and PTEN, kinases PDK1, mTORC2, and DNA-PK, and phosphatases PP2A phosphatases and PHLPP[21,22,37]. Despite both in vitro and in vivo results showing that PRMT5 inhibition markedly decreased AKT phosphorylation at Thr308 and Ser 473, none of the previously defined AKT activators were altered by the treatment (Fig. 3 and Supplementary Fig. 3). It is unlikely that PRMT5 activates AKT indirectly through an unidentified upstream regulator of AKT, although we cannot completely exclude this possibility. Nevertheless, we detected the association of PRMT5 with AKT (Fig. 4a). Further, we showed that reduced symmetric dimethylarginine on endogenous AKT1 but not AKT2 or AKT3 correlates with decreased phosphorylation under PRMT5 inhibition (Fig. 4b and Supplementary Fig. 4b, c). These results suggest that AKT1 is the primary target of PRMT5 in the context

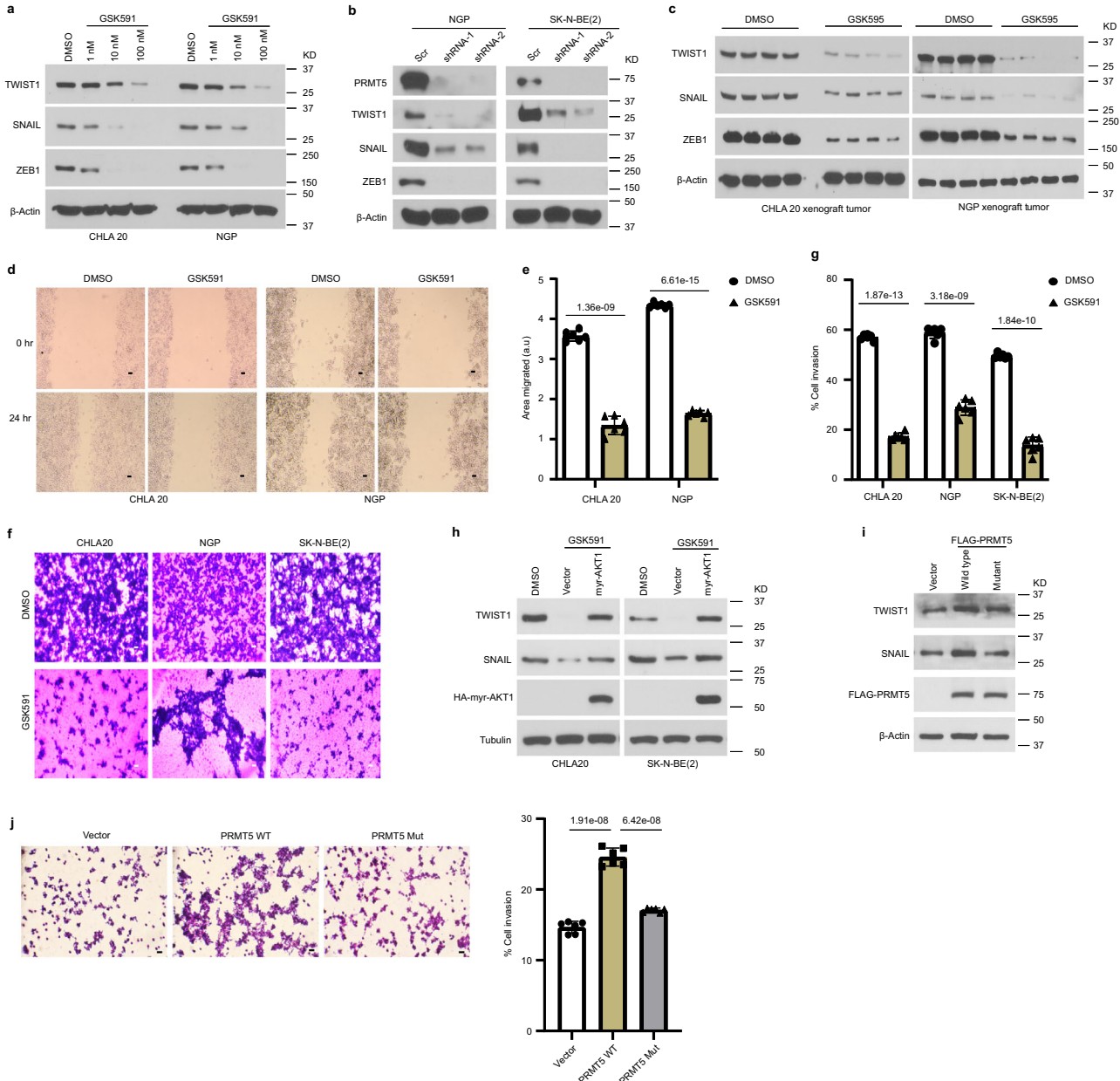

**Fig. 6 PRMT5/AKT regulates the EMT program. a** Immunoblots showing the expression of TWIST1, SNAIL, and ZEB1 in neuroblastoma cells treated with DMSO or increasing doses of GSK591 in CHLA20 and NGP cells. **b** Analysis of TWIST1, SNAIL, and ZEB1 protein levels in the scramble and shPRMT5 cells. **c** The expression of EMT transcription factor TWIST1, SNAIL, and ZEB1 was examined in xenograft tumors from mice treated with vehicle or GSK595 ($n = 4$). **d** Representative images of DMSO or GSK591-treated cells migrated to the cleared space (wound) after 24 h. Scale bars, 100 μm. **e** Quantification of in vitro cell migration assay ($n = 6$). The migration area was determined by measuring the total area of the wound using the ImageJ software. **f** Representative images of DMSO or GSK591-treated cells invaded to ECM coated membrane in transwell invasion assay. Scale bars, 100 μm. **g** Percentage of invasive cells normalized by cell numbers in the non-ECM coated 12-well plate using ImageJ ($n = 6$). **h** Protein levels of TWIST1 and SNAIL were analyzed in DMSO or GSK591-treated cells transfected with a vector or constitutively activated AKT1. **i** The protein levels of TWIST1 and SNAIL in cells overexpressing PRMT5 wild type or enzyme activity deficient mutant by Western blotting. **j** Representative images of CHLA20 cells overexpressing vector, wild type PRMT5, or an enzymatic deficient form of PRMT5 invaded to ECM coated membrane in the transwell invasion assay (left). Scale bars, 100 μm. Percentage of invasive cells normalized by cell numbers in the non-ECM coated 12-well plate using ImageJ ($n = 6$) (right). **e**, **g** *p* values were calculated by two-tailed unpaired Student's *t*-test using Microsoft Excel. **j** *p* values were determined using one-way ANOVA with Tukey's multiple comparisons test. Error bars represent SD. **a**, **b**, **d**–**j** Representative results from three independent experiments and the results shown are from a representative experiment. Uncropped immunoblots are provided in the Source Data file.

of neuroblastoma. We identified arginine 15 on AKT1 as the potential substrate of PRMT5, and confirm this by in vitro methylation assay (Fig. 4e). Remarkably, phosphorylation on Thr308 and Ser473 was diminished when AKT1-Arg 15 was mutated to lysine (Fig. 4f). These results provide substantial

evidence that PRMT5-mediated arginine methylation of AKT1 is necessary for its full activation (Fig. 4).

Another group described a role for PRMT5-mediated methylation of AKT1-R391 in breast cancer[46]. These authors showed that mutating AKT1-Arg 15 to lysine failed to impact AKT1

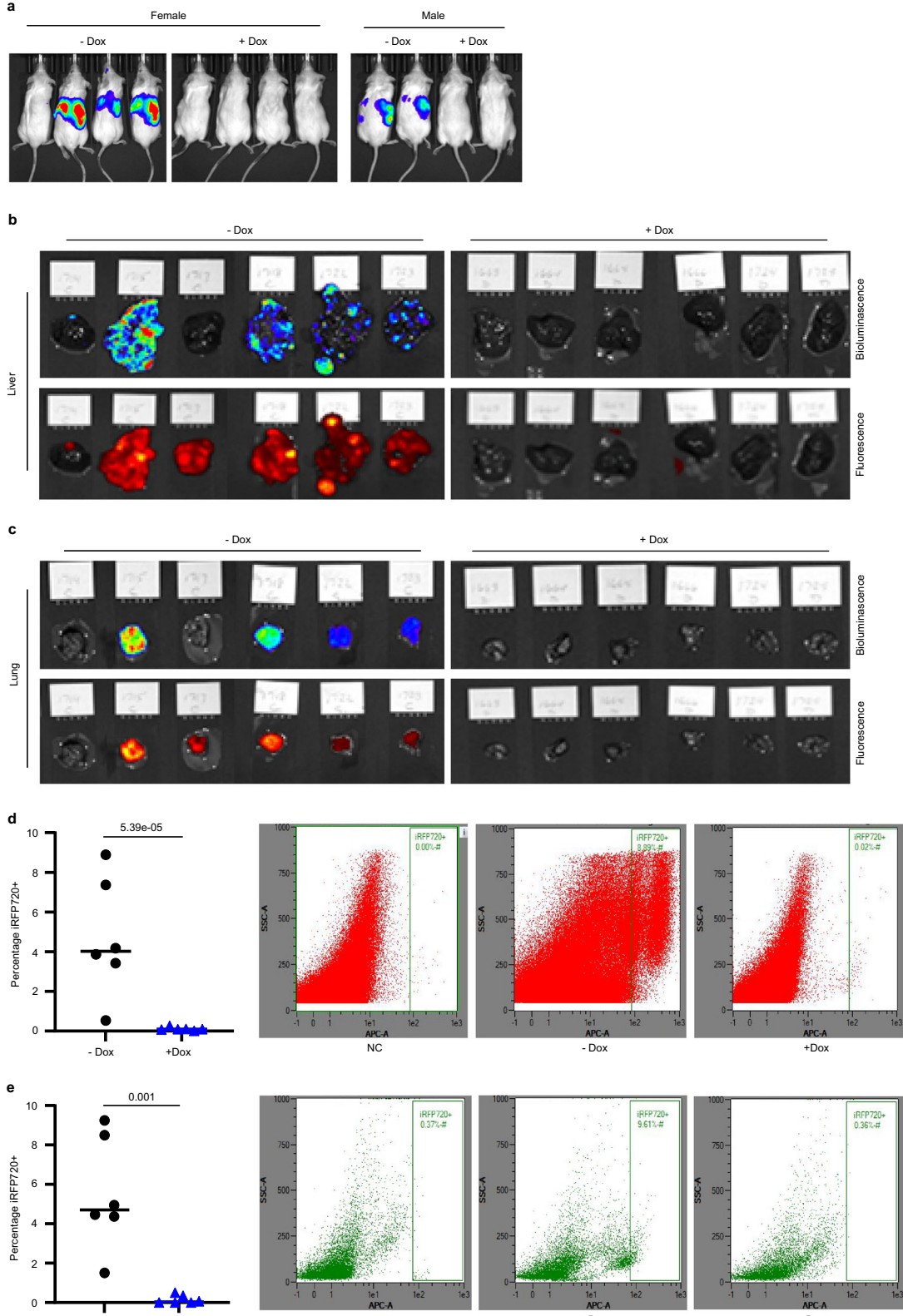

**Fig. 7 PRMT5 depletion effectively blocks tumor cell metastasis to the liver and lung. a** Bioluminescent imaging of mice tail vein injected SK-N-BE (2) shPRMT5 cells with or without doxycycline treatment ($n = 6$). **b** Bioluminescent (upper) and fluorescent imaging (lower) of livers harvested from mice described in **a**. **c** Bioluminescent (upper) and fluorescent imaging (lower) of lungs harvested from mice described in **a**. **d** FACS analysis of iRFP720+ human neuroblastoma cells in the liver from mice with or without doxycycline treatment ($n = 6$). **e** FACS analysis of iRFP720+ human neuroblastoma cells in the lung from mice with or without doxycycline treatment ($n = 6$). **d**, **e** $p$ values were determined by a two-tailed unpaired Student's $t$-test using Microsoft Excel. Error bars represent SD.

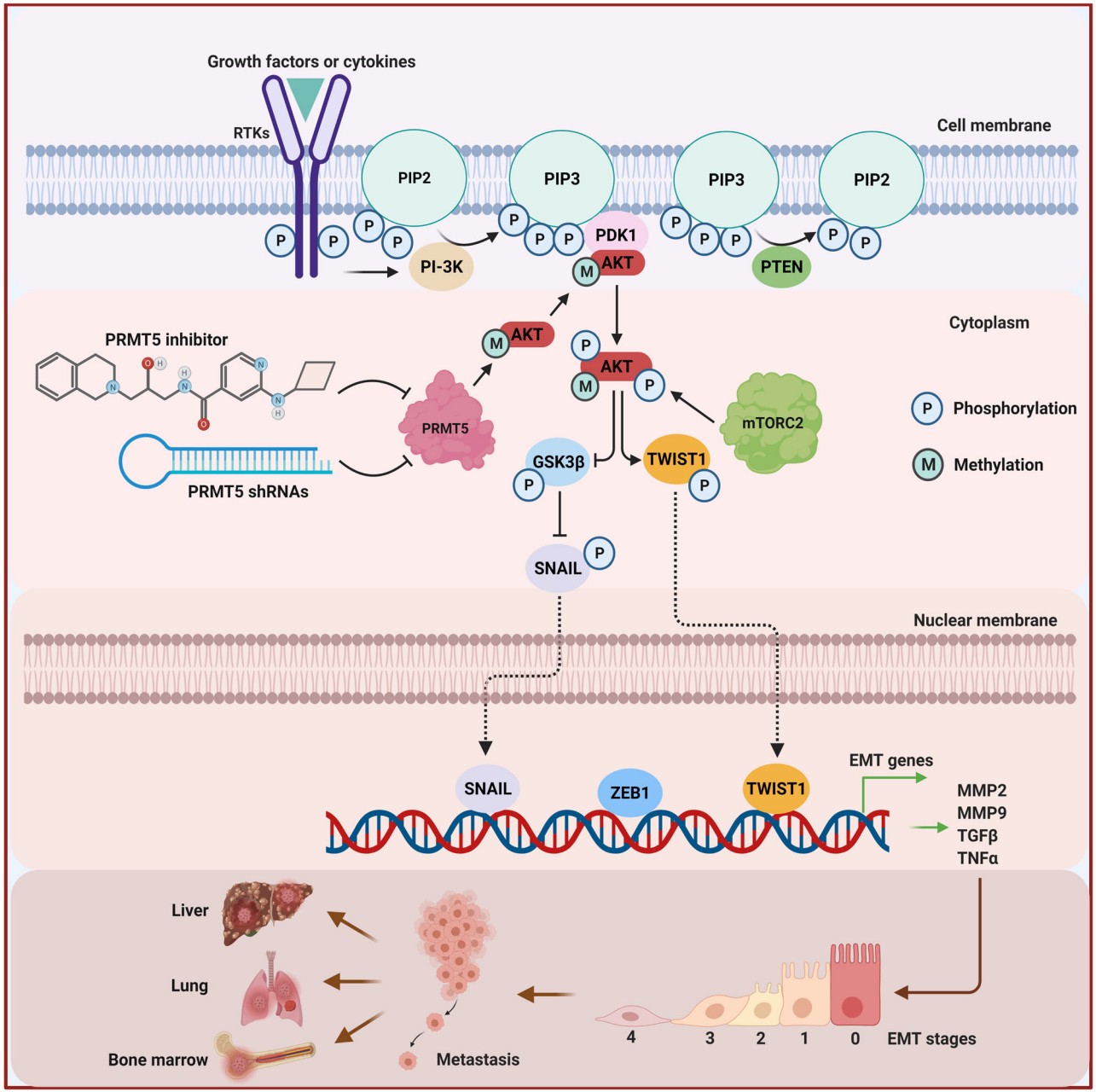

**Fig. 8 Graphic summary of the mechanism by which PRMT5 regulates AKT activation to promote metastasis.** AKT is activated in a cascade of events. Upstream stimuli, such as growth factors and cytokines, induce the production of phosphatidylinositol (3,4,5) trisphosphates (PIP3) by phosphoinositide 3-kinase (PI3K). These phospholipids serve as plasma membrane docking sites for AKT and PDK1 at pleckstrin-homology (PH) domains. PDK1 phosphorylates AKT at Thr308 at the plasma membrane which partially activates AKT, whereas AKT is fully activated after phosphorylation at Ser473 by mTORC2. PRMT5 methylates AKT1 on R15 in the PH domain, by which it promotes AKT1 association with the plasma membrane and subsequent phosphorylation by PDK1 and mTORC2. Downstream of AKT signaling, PRMT5 increases the expression of EMT transcription factors, such as SNAIL, ZEB1, and TWIST1 augmenting the EMT program to promote tumor metastasis.

SDMA in breast cancer lines, which contrasts with our findings in neuroblastoma. While testing the potential of AKT1-R391 being a substrate of PRMT5 in neuroblastoma, we found that the ectopic expression of AKT1-R391K mutant is significantly lower compared to AKT1 wild type and R15K mutant even in the presence of proteasome inhibitor MG132 (Supplementary Fig. 5e). Moreover, the molecular weight of AKT1-R391K is smaller than AKT1 wild type and R15K (Supplementary Fig. 5e). These results suggest that un-methylated R391 is unlikely a naturally occurring event in the context of neuroblastoma. Interestingly, these

authors did not detect significant changes of SDMA on AKT2/3 in breast cancer upon PRMT5 inhibition, which is consistent with our observation. We examined whether PRMT9, another type II protein methyltransferase generating symmetric dimethylarginine, is involved in the regulation of AKT activation. But there is no change in AKT phosphorylation when PRMT9 was reduced by siRNA-mediated gene silencing (Supplementary Fig. 4d). Together, our data highlight that PRMT5-mediated posttranslational modifications are highly context-dependent. A better understanding of the PRMT5-methylome in different cell types or

tissues may shed light on its specific function in many other cancer types and guide the precise design of therapeutic interventions.

AKT is a central node of signaling pathways deregulated in many human malignancies[22]. Therefore, it has been intensively pursued as an attractive drug target. Numerous AKT inhibitors were developed including non-selective ATP-competitive inhibitors, allosteric inhibitors, and irreversible inhibitors, all aiming at the kinase activity of AKT[47]. AKT consists of three domains, an amino-terminal pleckstrin-homology (PH) domain interacting with membrane lipids such as PIP3 and PIP2, a central kinase domain sharing similarity with AGC kinases, and carboxyl-terminal domain-containing regulatory elements. AKT activation includes three orchestrated events, a PH domain-dependent membrane translocation step, followed by the phosphorylation of the two key regulatory sites, Thr308 and Ser473[48]. We show loss of Arg 15 symmetric demethylation abolishes AKT1 binding to phosphatidylserine and relocation to the plasma membrane, and interaction with upstream kinases, leading to diminished phosphorylation of Thr308 and Ser473. Our results suggest an alternative strategy to develop inhibitory agents to block the oncogenic function of AKT.

Metastatic neuroblastoma represents a major clinical challenge, as less than 50% of children with high-risk aggressive cancer will survive[49]. Relapse of drug-resistant, metastatic disease remains the primary cause of death for these very young children. While AKT has diverse oncogenic roles, it also impacts chemotherapy sensitivity, modulates drug transporter expression, and contributes to drug resistance via PI3K/AKT/mTOR dependent signaling[50]. Thus, AKT has been the focus of multiple targeting efforts, including allosteric, ATP-binding pocket and more recently PROTAC mediated inhibition strategies[51]. Indeed, pan-AKT inhibition sensitizes neuroblastoma and other cancers to chemotherapy[52], and earlier clinical trials have tested anti-AKT therapeutics for neuroblastoma[53]. However, toxicity and resistance to AKT enzymatic activity via mutation has limited these approaches[48,54]. Our results reveal that pharmacologic or genetic PRMT5 inhibition blocks the expression of SNAIL, TWIST, and ZEB1 and almost completely blocks metastases in aggressive in vivo tumor models. It is noteworthy that constitutively activated AKT1 is sufficient to restore the decrease of these EMT transcription factors (Fig. 6h). Based on our findings we propose that targeting PRMT5 will limit AKT1 activation, metastases, and drug resistance of neuroblastoma and other aggressive cancers. We further suggest that combination therapies targeting both PRMT5 and AKT pathways may have synergistic impacts in vivo and warrant further preclinical and clinical investigation.

## Methods

**Chemicals**. PRMT5 methyltransferase activity inhibitor GSK3203591 was purchased from Cayman Chemical (cat#1616391-87-7). The in vivo active sister compound GSK3326595 was purchased from Chemietek (cat#CT-GSK332). Doxycycline hyclate was purchased from Sigma-Aldrich (cat#D9891). All chemicals were dissolved in 100% DMSO as stock solution and subsequently diluted in phosphate-buffered saline or growth media to working solution.

**Cell culture**. CHLA20 and SK-N-BE (2) cell lines were obtained from The Children's Oncology Group. CHLA20 cells were cultured in IMEM medium (Invitrogen, San Diego, CA, USA) supplemented with 10% fetal bovine serum (FBS) (Sigma-Aldrich, St. Louis, MO), 100 µg/ml streptomycin (Gibco), and ITS (5 µg/mL insulin, 5 µg/mL transferrin, and 5 ng/ml sodium selenite, Invitrogen). SK-N-BE (2) and NGP cells were grown in RPMI1640 medium (Invitrogen) supplemented with 10% FBS (Sigma-Aldrich), 100 µg/ml streptomycin.

**Plasmids**. Human pCDNA3/Flag-HA-AKT1, Plncx/myr-HA-AKT1, pCDNA6A/EGFR, and pHIV-iRFP720-E2A-Luc were purchased from Addgene. pCDNA3.1/Flag-PRMT5 wild type and dead enzyme mutant were as described previously[55]. pCDNA3/Flag-HA-AKT1-R15K mutant was generated by site mutagenesis using

QuickChange Lightning Site-Directed Mutagenesis Kit (Agilent, cat#210518) following the manufacturer's protocol. Primers introducing mutation sites were designed by Agilent PrimerDesign Program on the Agilent website, AKT1-R15K_forward 5′-GATGTACTCCCCTTTTTTGTGCAGCCAACCCTCCTT-CACA-3′, AKT1-R15K_reverse 5′-TGTGAAGGAGGGTTGGCTGCA-CAAAAAGGGGAGTACATC-3′. pLKO-Tet-ON inducible shPRMT5 constructs were generous gifts from Dr. William R. Sellers (Broad Institute). siRNA targeting PRMT9 was purchased from Thermo Fisher Scientific Inc. (cat#4392420).

**Stable cell lines**. CHLA20 and NGP cells were transduced with pHIV-iRFP720-E2A-Luc and enriched by FACS sorting on RFP 700 channels. NGP and SK-N-BE (2) cells were transduced with pLKO-Tet-ON inducible scramble or shRNA targeting PRMT5 followed by puromycin selection[45].

**Fluorescent activated cell sorting (FACS)**. Primary cells were isolated from xenograft tumors and liver. The percentage of iRFP720 positive cells was analyzed by FACS either on BD LSRII using Flow Jo 10.7.1 or Miltenyi MACSQuant 10 cytometers using MACSQuantifi 2.13.3 software.

**Cell viability assay**. Cells were seeded in 96-well plates and then maintained in the presence of vehicle or increasing doses of GSK591 for 6 days before the addition of 20 µL CellTiter 96 AQueous One Solution per well (Promega Corporation, Madison, WI. cat#G3582). Plates were incubated for 2 h before absorbance at OD490 was measured with a Synergy H4 Hybrid microplate reader (Bio Tek, Winooski, VT) using Gen5 software. Wells containing medium only were used as background for the measurement.

**Hoechst 33342 staining**. The cells were treated with DMSO or GSK591 100 nM for 6 days and then were stained with Hoechst 33342 (Thermo Fisher Inc., cat#H3570) for 10 min at 37 °C. The cells were photographed at 40× magnifications with fluorescence microscopy from Carl Zeiss Microscopy. All images were captured on an AXIO Observer microscope with a ZEISS Axiocam 506 mono camera and Zen 2 Pro software.

**Caspase-3/7 staining**. Apoptosis was measured by caspase-3/7 staining using CellEvent® Caspase-3/7 Green ReadyProbes® Reagent (Thermo Fisher Scientific Inc., cat# R37111). CHLA20 and NGP cells were grown until confluence on six-well plates with 6 days of treatment with DMSO or GSK591. Cells were fixed in 4% formaldehyde for 20 min and followed by methanol 100% for 10 min. Cells were stained with two drops of Caspase-3/7 green reagent per mL of media for 30 min at room temperature, and the nuclei were stained with DAPI (Sigma-Aldrich, cat#D9542) for 15 min in PBS. The images were taken under a Zeiss microscope with a 10× objective. The Caspase-3/7 activity was determined by analyzing particles using the ImageJ software. The percentage of Caspase-3/7 positive cells was normalized to the nuclei stained by DAPI.

**Xenograft mouse models**. All procedures were approved by the Institutional Animal Care and Utilization Committee (IACUC) at University of Massachusetts Chan Medical School. The maximal tumor size/burden permitted in our IACUC protocol is 1.5 cubic centimeters. Renal capsule implantation: NOD/SCID mice from Jackson Laboratory (NOD. Cg-Prkdc^scid/J, Strain#00130) were used in this study. One million iRFP720-luciferase transduced CHLA20 or NGP cells suspended in 0.1 ml of PBS were surgically implanted in the left renal capsule of mice, both male and female, at the age of 8 weeks. Tumor growth was monitored weekly by bioluminescent imaging (IVIS Lumina XR System, Caliper Life Sciences, Hopkinton, MA, USA). After 10 days, mice bearing tumors with similar sizes (determined by live animal imaging of bioluminescence) were randomly divided into either a "vehicle control" group (0.5 methylcellulose, Sigma-Aldrich #M0430) or a PRMT5 inhibitor "GSK595 treatment" group (100 mg/kg). Animals were treated for 2 weeks, with 100 mg/kg GSK595 or vehicle by oral gavage twice daily.

Tail vein injection: One million cells were injected via tail vein into NOD/SCID mice, both male and female, at the age of 8 weeks. Then the mice were randomized into two groups, one treated with water containing 1 mg/mL doxycycline which induces the expression of shRNA against PRMT5, and the other on control water. After 10 weeks, mice were imaged, euthanized, and organs were harvested for analysis. The body weight of mice was monitored weekly. At the end of the treatment, all mice were euthanized. Tumors and the right kidneys (control) were dissected and weighed. Tumor, kidney, and liver were subjected to organ imaging by bioluminescence. Live animals and ex vivo organs were imaged in IVIS Imaging System using Caliper Life Sciences Living Image® Software Version 3.2; ex vivo organs were photographed using iPhone camera.

**Western blot**. Tissue or cells were washed with cold PBS and lysed in 1% NP-40 lysis buffer containing a protease and phosphatase inhibitor cocktail (Roche Diagnostics, Indianapolis, IN, USA). Proteins were analyzed by SDS-PAGE, followed by immunoblotting with indicated primary antibodies. Blots were incubated with horseradish peroxidase-conjugated anti-mouse or anti-rabbit IgG antibodies, and proteins were detected by enhanced chemiluminescence (Thermo Fisher

Scientific Inc., cat#7074). Antibodies used in this study were listed in Supplementary Table 1.

**Immunofluorescence**. Cells were treated with DMSO or GSK591 100 nM for 6 days and fixed with 4% paraformaldehyde for 20 min at room temperature, followed by permeabilization in ice-cold methanol for 10 min at −20 °C. Cells were then incubated in blocking buffer (5% BSA in PBS) for 1 h at room temperature, followed by incubation with anti-PRMT5 antibody, anti-Thr308-AKT, anti-Ser473-AKT, and total-AKT antibody at 4 °C overnight. Cells were washed three times for 5 min in PBS, then incubated for 2 h with Alexa Fluor 488-conjugated goat anti-rabbit secondary antibody against phospho-AKT and total-AKT, and Alexa Fluor 594-conjugated goat anti-mouse secondary antibody for PRMT5 (diluted 1:100 in blocking buffer) at room temperature. Finally, the nuclei were stained with DAPI (Sigma-Aldrich, cat#10236276001) for 30 min at room temperature before visualization. Cells were observed with ZEISS Axiocam 506 mono Digital Camera for Fluorescence Microscopy.

**Confocal microscopy**. CHLA20 cells were transfected with vector, HA-tagged AKT1 wild type or R15K mutant. Twenty-four-hour post-transfection, the cells were serum-starved overnight and stimulated with or without EGF (10 ng/ml) for 30 min. Next, the cells were fixed with 4% formalin at room temperature for 20 min, gently rinsed twice with PBS, then permeabilized with ice-cold 100% methanol at −20 °C for 10 min. Subsequently, the cells were incubated with blocking buffer (PBS containing 5% normal goat serum and 0.3% Triton X-100) for 60 min at room temperature and incubated with HA antibody (1:500 dilution) and β-actin antibody (1:500 dilution) in blocking buffer at 4 °C overnight. After incubation, the cells were gently washed twice with PBS and incubated with Alexa Fluor 488-conjugated and 594-conjugated secondary antibodies at room temperature for 2 h. Finally, the cells were stained with DAPI at room temperature for 30 min, and images were acquired with an inverted Nikon Eclipse Ti2 confocal microscope (Nikon Instruments/Nikon Corp).

**Immunoprecipitation (IP)**. CHLA20 and SK-N-BE (2) cells were seeded in 10 cm culture dishes. Proteins were extracted from those cells using lysis buffer (20 mM Tris, PH 7.4, 150 mM NaCl, 2 mM EDTA, 2 mM EGTA, 1 mM sodium orthovanadate, 50 mM sodium fluoride, 1% Triton X-100, 0.1% SDS, and 100 mM phenylmethylsulfonyl fluoride) and centrifuged at $16,000 \times g$ for 10 min at 4 °C. The cell lysates were pre-cleared with protein A or protein G beads (Santa Cruz Biotechnology, cat#sc-20001, cat#sc-2002) at 4 °C for 1 h. Subsequently, 5 μg primary antibody or isotype IgG was added to the cleared cell extracts and incubated at 4 °C overnight. Protein A or protein G beads were added to the cell extracts and incubated at 4 °C for 3 h. Then the beads were washed three times with wash buffer (100 mM NaCl, 50 mM Tris pH 7.5, 0.1% NP-40, 3% glycerol, and 100 mM phenylmethylsulfonyl fluoride). Finally, the beads-bound proteins were eluted by adding 2x Laemmli sample buffer and heated at 95 °C for 5 min. Eluted proteins were analyzed by western blotting.

**In vitro methylation assay**. The methylation of AKT1 wild type or R15K was examined as previously reported[56]. Briefly, AKT1 wild type or R15K was transfected to CHLA20 cells and HA-tagged proteins were precipitated by HA beads. The precipitated proteins on HA beads were incubated with 0.5 μCi $S$-adenosyl-L-[methyl-$^{14}$C] (Perkin–Elmer, cat#NEC363010UC) with or without 0.5 μg recombinant PRMT5/MEP50 (Sigma-Aldrich, cat#SRP0146) for 1 h at 30 °C. The reaction was stopped by the addition of 2x Laemmli buffer (Biorad) containing 0.2 M dithiothreitol (DTT) and heated at 95 °C for 5 min. The eluted proteins were separated from beads on a magnetic stand. 20 μL eluted proteins were run on SDS-PAGE gel and transferred to the PVDF membrane. $^{14}$C signal was visualized by exposure the membrane to high sensitivity X-ray film. Eluted proteins were run on SDS-PAGE gel and transferred to the PVDF membrane.

**Phosphatidylserine binding assay**. pCDNA HA-FLAG-AKT1 wild type or R15K mutant were transfected to CHLA20 cells. Twenty-four hours post-transfection, cells were lysed in 1X RIPA buffer plus protease and phosphatase inhibitors (Roche, cat#a32961). HA-tagged proteins were pulled down by anti-HA magnetic beads (Thermo Fisher Scientific Inc., cat#88837) and eluted from beads by incubation for 15 min at 37 °C with 4x column volume of HA synthetic peptide solution (1 mg/mL in TBS) (Thermo Fisher Scientific Inc., cat#26184). The eluted proteins were incubated with 30 ul of phosphatidylserine-agarose beads (Echelon Biosciences cat#P-B0PS) for 1 h at 4 °C in the binding buffer containing 10 mM HEPES (pH 7.4), 150 mM NaCl, and 0.5% IGEPAL CA630 (Sigma-Aldrich, cat#56741). The beads were washed three times with binding buffer and eluted by boiling 5 min in 1X SDS gel loading buffer. Eluted proteins were run on SDS-PAGE gel and transferred to the PVDF membrane. The membrane was incubated with an anti-HA antibody to detect HA-tagged proteins.

**Bioinformatics analysis**. The gene expression and clinical data from 461 neuroblastoma patients were downloaded from the GEO database GSE49711 RNA-seq data series (https://www.ncbi.nlm.nih.gov/geo/query/acc.cgi?acc=GSE49711).

Three sets of microarray data from neuroblastoma patient cohorts were downloaded from the R2 database (R2: Genomics analysis and visualization platform (http://r2.amc.nl). The relative expression of PRMT5 and survival analysis was performed on these two groups using the survminer R package.

**ActivSignal IPAD assay**. CHLA20 and NGP cells were treated with DMSO or 100 nM GSK591 for 6 days, harvested, and cell pellets were snap-frozen in liquid $N_2$. Then whole-cell lysate was submitted to ActivSignal for analyzing the expression or phosphorylation of 70 pivotal proteins involved in more than 20 signaling pathways. The signal was normalized to the expression of housekeeping genes. Each pathway was covered by multiple targets. Samples were analyzed in independent biological duplicates.

**Cell migration assay**. In vitro cell migration assay was assessed by wound healing. CHLA20 and NGP cells were grown until confluence in six-well plates with 6 days of treatment of DMSO or 100 nM GSK591. Then cells were incubated for 2 h with 10 μg/ml mitomycin C (Sigma-Aldrich, cat#m5353) to inhibit cell proliferation before the wound generation. Cells were maintained in media containing 10 μg/ml mitomycin C afterward. A sterilized 200 μl pipette tip was used to generate wounds across the cell monolayer, and debris was removed by washing with media. The monolayers were then incubated for 24 h at 37 °C. The progress of cell migration into the wound was monitored using a Zeiss microscope with a 10× objective. The bottom of the plate was marked for reference, and the same field of the monolayers was photographed at 0 and 24 h. Five images per sample were analyzed. The distance between the edges of the wound was measured at times 0 and 24 h, and the reported migrated distance corresponds to the difference between these two. The migration area was determined by measuring the total area of the wound using the ImageJ software.

**Cell invasion assay**. In vitro invasion assay was performed in cell culture inserts with 8 μm pore size (Corning Life Science, cat#354578) coated with extracellular matrix (ECM) from Engelbreth–Holm–Swarm murine sarcoma (Millipore Sigma, cat#E1270). The basement membrane of the inserts was coated with a 50 μL of ECM gel at room temperature, then hydrolyzed at 37 °C for 30 min. CHLA20, NGP, and SK-N-BE(2) were treated with either DMSO or 100 nM GSK591 for 6 days, then detached and suspended at $5 \times 10^4$ cells/ml in serum-free media containing DMSO or 100 nM GSK591, respectively. Two hundred microliter cell suspension was added onto the upper surface of the insert. Seven hundred and fifty microliter growth media were added to the bottom chamber. After 15 h incubation, cells grown on the top surface of the insert were removed with cotton swabs. To examine the impact of PRMT5 overexpression in cells, the incubation time was shortened to 8 h. Cells that invaded the opposite surface of the insert were washed with PBB, fixed in 4% formaldehyde for 5 min, and followed by methanol 100% for 20 min, then stained with 0.25% crystal violet for 10 min. Photographs were taken under a light microscope and quantified using ImageJ software. The percentage of cells of invasion was normalized to the equal number of cells plated in a 24-well plate cultured for the same period.

**Statistical analyses**. All quantitative data points represent the mean of three independent experiments performed in triplicates with standard deviation (S.D). Unless indicated in the figure legend, statistical analysis was performed using one-way ANOVA (GraphPad Software, Inc., La Jolla, CA) or unpaired $t$-test (Microsoft Excel). The IC50 value was determined by nonlinear regression (curve fit) using the log (inhibitor) versus response (three parameters) model.

**Reporting summary**. Further information on research design is available in the Nature Research Reporting Summary linked to this article.

## Data availability

Three sets of microarray data from neuroblastoma patient cohorts were downloaded from open access R2 database (R2: Genomics analysis and visualization platform [http://r2.amc.nl]. Publicly available GSE49711 RNA-seq data series presented in Fig. 1a were downloaded from NCBI GEO [https://www.ncbi.nlm.nih.gov/geo/query/acc.cgi?acc=GSE49711]. All additional data is included in the Supplementary Information and Source Data files. Source data are provided with this paper.

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

## Acknowledgements

We thank Drs. Thomas Fazzio, Heidi Tissenbaum, Anthony Imbalzano, Jeffrey Nickerson, Ivana de la Serna, and Michael Green for their invaluable insights to improve this manuscript. This work is supported by the Dean's Research Fund (University of Massachusetts Chan Medical School) (M.M.L. and Q.W.) and Hyundai Scholar Hope Grant (Hyundai Hope on Wheels Foundation) (J.M.S.).

## Author contributions

L.H., E.J.R., X.S., O.V.-T., and K.W. designed and performed the experiments, and analyzed the data. X.-O.Z. and B.S. performed bioinformatic analyses. T.H. and J.P.C. provided reagents. D.M., G.W., and L.Z. performed the experiments. M.M.L. and J.M.S. designed the experiments and analyzed the data. Q.W. conceived the project, designed and performed experiments, analyzed the data, and prepared the manuscript.

## Competing interests

The authors declare no competing interests.

**Additional information**

