## [Peer Review File · Nature Communications]

REVIEWERS' COMMENTS

Reviewer #1 (Remarks to the Author):

The manuscript by Lei Huang et al present a novel function of PRMT5 that regulates AKT and EGFR in neuroblastoma cells. Specifically, methylation of ARG 15 on AKT-1 is required for its phosphorylation in the activation motifs (S473 and T308). In addition, inhibition of PRMT5 causes downregulation of EGFR expression, and the two biochemical effects were found to be independent of each other. Finally, PRMT5 enhances cell growth and EMT and as expected, inhibition of PRMT5 leads to reduction in cell growth, invasion and metastasis in vitro and in xenograft models. Although interesting and novel, there are quite a few problems with the manuscript, including proper controls for IP, lack of studies on stimulated AKT phosphorylation, improbable blots, and lack of quantification that should be addressed prior to publication. The detailed comments are as follows:

- 1) In Fig. 1a, comparing the expression of PRMT5 between stage 4 and stage 3 neuroblastoma cancer: 126 Patients in stage 4 and only 6 patients in stage 3? The unequal numbers seem to drastically divert the statistics. I would suggest replacing this panel by a more indicative data.
- 2) It is suggested that methylation of AKT-1 by PRMT5 is a pre-requisite for AKT phosphorylation but how this is happening is neither explored nor discussed. It would be interesting to test if methylation of AKT-1 is demanded for the anchorage of AKT in the plasma membranes, where it is normally phosphorylated. Also, the paper can benefit from showing actual methylation of the protein using mass spectroscopy.
- 3) In figure 3, it is possible that at least some of the reduced AKT phosphorylation is due to effect of the RPMT5 inhibitor on upstream components including RTKs. The heatmap in Fig.3A shows effects on MAPKs and their downstream targets (e.g. p53-Ser15 or pRSK), but no downstream targets of AKT. These point is not addressed in the following figures as well. It is recommended to follow the phosphorylation of known AKT substrates such as GSK3 or Foxo1 in the presence of the PRMT-5 inhibitors to verify that the methylation is indeed required for AKT activity.
- 4) The results represent in figure 3 demonstrate the effect of PRMT5 on the basal phosphorylation of AKT, but the effect of stimulated phosphorylation is ignored. Due to the effects on the receptors, it is important to show what is the effect of EGF/FGF/NGF stimulation on the PRMT-5 effects.
- 5) In Fig. 4 and Fig S3, the immunoprecipitation (IP) and Co-IP of the AKT isoforms is not demonstrated. The authors should present proper controls for each of the IPs in order to verify that the signals are indeed from the specified isoforms with no cross reactivity between them. This will also verify that the amounts precipitated were equal. Also, a verification of the Ab specificity of each isoform should be done (by siRNA) in a separate experiment. As expected, there are changes in the molecular weight of the different AKT isoforms when

tested in the CHLA 20 cells. However, no changes are seen in NGP cells, and I wonder if this is reproducible or what might be the reason for it. Importantly, it is not clear why the IgG band is not apparent in any of the blots from the Co-IP experiments.

6) In continuation to point 5, the amount of precipitated proteins should be shown and quantitated in all IP and CoIP experiments. Input controls are not sufficient in this case.

7) The legend of Fig. 4 is truncated, and therefore the details of the experiments of Fig. 4f are not known. Moreover, the amount of IPed AKT1 is not presented, and the phosphorylation is of AKTs from unstimulated cells. Finally, from what I understand, it is still possible that the effect on AKT1 phosphorylation is not direct, and might be affected by interaction with regulators or affect EGFR expression (as in Fig. 7). This is an essential point that should be nailed by the authors; for example by using an in-vitro phosphorylation with active PDK1, and by examining the role of the mutation on AKT binding to other proteins (interactome). Two other points: it is not clear why the IgG band is not seen in the IP experiment, and why do the authors use the term AKT1 15G/K, 15R/K, or better, R15K, as appears in page 9. Finally, the reduction of SMDA of AKT-1 by GSK591 (Fig. 4C) does not look significant.

8) In Fig 7, it seems that GSK591 affects all RTKs, indicating that the effect might be global and may affect the degradation or trafficking of the receptors. This point should be better explained. In addition, is there a reasonable explanation as to why the expression of EGFR (in the presence of GSK591) is dramatically reduced in CHLA 20 cells and only moderately in NGP cells? Also, in another experiment (Fig 7E), the effect of GSK591 on EGFR expression in CHLA 20 is also moderate. Which of the two experiments represents the real effect of GSK591? Quantification of the blot should be helpful. Finally, it is not clear to me why MK-2206 affects AKT phosphorylation. It is an AKT inhibitor and is not supposed to inhibit the PDK1 or mTORC2 activity, and indeed we and other colleagues do not detect such a change.

9) In many cases, the presented data are either not probable or not explained well enough. For example, why don't the authors detect the IgG in the IP experiments, What is the reason for the big difference between the results in Fig. S1a and S1c in the control lanes. The background in all experiments is too homogeneous, which may indicate that the figures were heavily manipulated. Therefore, I suggest to provide the original blots in the supplementary. Finally the molecular weight markers are problematic and not reproducible in many cases. For example, PRMT5 appears in some cases in ~73 kDa, but in other blots (e.g. Fig. 6C) at ~76 kDa. The molecular weight of AKT isoforms should be different between each other. PP2A-C is a 34 kDa protein (not 39 as appear in Fig. S3b), the MW of PHLPP1 and 2 seem to be replaced, PP2A-A is 65 kDa and it is not clear what PP2A B is used.

10) Several minor points are: a) in Figure 7D the effect of the phosphorylation of EGFR by WT or mutant AKT-1 is not shown (only the expression is shown). b) in Figure 4B: Based on the results and in contrast to what is written (page 8, rows: 8-9), the interaction between PRMT5 and AKT (or pAKT) does not change following inhibition of PRMT5. Rather,

GSK591 “somehow” interfere with the phosphorylation of AKT (by PDK-1 and mTORC-2). This point should be explained. c) Do the authors know if PRMT5 plays a role in AKT activation in different cell lines as well? How general is the role of PRMT5 in AKT activation? d) In the introduction, the explanation of the way AKT is activated is not accurate, and one of the kinases that phosphorylates Ser473 is mTORC2 (not rictor).

Reviewer #2 (Remarks to the Author):

This paper identifies a role for PRMT5 in the neuroblastoma (NB) tumour growth and metastatic dissemination, and identifies Akt as a novel PRMT5 substrate. In parallel, they find that PRMT5 regulates the expression of growth factor receptors, including EGFR, possibility through epigenetic-mediated regulation. They conclude that PRMT5 promotes NB metastatic growth through independent and Akt and EGFR pathways.

Many of the concepts in this study are not novel. A number of papers have already linked PRMT5 to the phosphorylation and activation of AKT (Zhang et al., J Cell Mol Med, 2019; Chung et al., JBC, 2019; Strobl et al., Mol. Can. Therapeutics, 2020; Li et al., Cell Transplant. 2019)), knockdown of PRMT5 has already been shown to promote apoptosis in SK-N-BE2 cells (Park et al., Molecular Oncology 2015) and, as pointed out by the authors, PRMT5 already has an established role in metastatic tumour cell growth. Yet, the *in vivo* data demonstrating the potency of GSK591 on NB xenografts growth and liver metastasis is compelling, and the use of these tumours to demonstrate *in vivo* relevance of *in vitro* cell culture findings throughout the study is a strong addition.

Unfortunately, I do not find that the overall conclusions regarding mechanistic insight are fully supported by data presented. In line with this, the addition of the rather weak EGFR epigenetic data distracts from the main interesting finding regarding how methylation promotes Akt activation, and feel that if the authors had focused on the latter, the paper would have been more compelling.

Comments:

It appears that administration of GSK595 completely blocks tumour growth, so much so that data presented in Fig 2b and c demonstrate that there is no tumour mass (although there must be something as downstream analysis was conducted). The authors state that GSK595 was administered when tumours reached a certain luciferase pixel intensity, however as this is an arbitrary number it is difficult to know how big tumours were at the start of treatment. I am wondering if the authors observed tumour regression? It would be nice to see data of luciferase intensity at the start of GSK595 treatment and at the end. I am also wondering why the *in vivo* experiment was not conducted using Sk-N-BE2 cells as this was derived from a relapsed tumour. If GSK595 has an effect in this cell line, this could have significant clinical implications for second line therapy options. What happens to tumour growth if GSK595 is administered after tumours have grown to a well-established size?

I find the colour scale of Fig 3a rather confusing. I am assuming that green indicates a decrease in protein expression/PTM, however the scale goes from 0-9. Most of the targets are a white bar, which I am assuming that this means no change after DMSO treatment. Shouldn't this be zero?

At the start of the "PRMT5 methylates AKT" section, the authors list a number of proteins involved in AKT signalling. Are any of these genes mutated in the cell lines used? Are the basal levels of AKT signalling increased compared to non-transformed cells? Following on from this, is PRMT5 required for growth factor mediated AKT signalling? This will enable a distinction between PRMT5 regulating basal or induced AKT signalling. Likewise, it may be that proteins blotted for in Supp Fig 3 are PRMT5 dependent after stimulation.

The finding that GSK591 suppresses Thr308 and 473 phosphorylation should be substantiated by knockdown analysis.

Why does GSK591 not reduce the apparent SDMA on AKT2/3? How convinced are the authors on the specificity of their SDMA antibody (I could not find details of it the antibody table). Are AKT2/3 methylated by PRMT9? Experiments should to be repeated using shPRMT5 and in vitro methylation assays with recombinant proteins.

Data in Figure 4b and accompanying supplementary is not convincing. The authors should use PLA for co-localisation analysis.

In many figures, after immunoprecipitation the western blot of protein used as bait is omitted (e.g. Fig 4a). This must be included.

If PRMT5 is required for Thr308/473 phosphorylation, one would expect reduced phosphorylation of downstream substrates. Why was this not shown?

A major criticism of this paper is that the mechanism by which AKT methylation leads to phosphorylation has not been demonstrated. Does methylation recruit activating kinase?

The second major criticism is the inclusion of the RNA-seq and EGFR promoter analysis: it feels like the authors had two separate stories that they have tried to bring together. More importantly, data presenting is not convincing. Why were two different cell lines use for 6f and g? Why was IgG control ChIP not included? Why was H3R3me2s not examined, particularly as this is the recognised PRMT5 epigenetic mark that promotes gene expression, not H4R3me2s, which generally correlates with gene repression? Why was EGFR activation not specially looked at rather than the unconvincing data in Fig 6e?

Fig 7A - I do not understand the rationale of ectopic PRMT5 expression in GSK591 expressing cells. Surely it makes more sense to do the experiment in shPRMT5 cells?

Figure legends are difficult to follow. Most of the times the signpost letter is placed at the end of the sentence rather than the front, and then it flips.

Minor points:

Can more information be provided in the text regarding the proteomics-based target screen in Fig 3?

Gene identified in the overlap of the two cell lines should be listed (Fig 5b).

Catalogue number for H3R8me2s and H4R3me2s repeated twice in table.

Not sure what the “dimer EGFR” and arrow to “cancer Cells” in the schematic signifies in relationship to this study.

Figure legend 3: typo – full cell name for BE2 omitted.

Reviewer #3 (Remarks to the Author):

These authors show that PRMT5 promotes neuroblastoma metastasis by increasing the transcription of EGFR and elevating AKT1 activation by methylation on Arg 151. I thought the biochemistry was sound, however the mouse in vivo studies could be improved.

Specifically, can survival curves be added for Fig 2. Can this compound be analyzed in PDX models in addition to the single cell line xenograft shown? The data that PRMT5 is a driver of metastases is not particularly strong. Can metastases be examined also in PDX studies in vivo? Do levels of PRMT5 correlate with metastases specifically in human tumor databases, and does introduction of PRMT5 into cell lines and PDXs drive metastases?

2. In western blots in Fig 3, were cells examined at steady state? Was serum or were growth factors added prior to lysis? Supplemental Fig 3, although a negative result, is a very convincing and clean figure.

3. I am not sure I understand the rescue experiments in Fig 7. If you are treating cells with inhibitor, then why doesn't inhibitor block the activity of the transfected PRMT5 in addition to the endogenous protein?

Point-by-Point Response to Reviewer's Comments

REVIEWER COMMENTS

We thank Reviewer#1 for recognizing the novelty of our study and pointing out some technical issues to help us improve this manuscript.

Reviewer #1 (Remarks to the Author):

The manuscript by Lei Huang et al present a novel function of PRMT5 that regulates AKT and EGFR in neuroblastoma cells. Specifically, methylation of ARG 15 on AKT-1 is required for its phosphorylation in the activation motifs (S473 and T308). In addition, inhibition of PRMT5 causes downregulation of EGFR expression, and the two biochemical effects were found to be independent of each other. Finally, PRMT5 enhances cell growth and EMT and as expected, inhibition of PRMT5 leads to reduction in cell growth, invasion and metastasis in vitro and in xenograft models. Although interesting and novel, there are quite a few problems with the manuscript, including proper controls for IP, lack of studies on stimulated AKT phosphorylation, improbable blots, and lack of quantification that should be addressed prior to publication. The detailed comments are as follows:

1) In Fig. 1a, comparing the expression of PRMT5 between stage 4 and stage 3 neuroblastoma cancer: 126 Patients in stage 4 and only 6 patients in stage 3? The unequal numbers seem to drastically divert the statistics. I would suggest replacing this panel by a more indicative data.

Response: We agree with this concern. The RNA-seq data in original Figure 1a were extracted from the TARGET database, which has a limited number of patients. In the revised manuscript, we have addressed this directly by including a new analysis of RNA-seq datasets from a large cohort of neuroblastoma patients from a paper published in Genome Biology (Genome Biol. 2015. 16(1):133. PMID: 26109056) (revised Fig. 1a). The conclusions are unchanged with the new datasets.

2) It is suggested that methylation of AKT-1 by PRMT5 is a pre-requisite for AKT phosphorylation but how this is happening is neither explored nor discussed. It would be interesting to test if methylation of AKT-1 is demanded for the anchorage of AKT in the plasma membranes, where it is normally phosphorylated. Also, the paper can benefit from showing actual methylation of the protein using mass spectroscopy.

Response: We thank the reviewer for prompting us to strengthen this point. In the revised manuscript, we have included new data where we examined the association of AKT1 wild type or R15K mutant with the plasma membrane in the following experiments. 1) We separated the plasma membrane fraction from the cytosolic fraction and examined the levels of phospho-AKT1 in GSK591 treated cells and PRMT5 knockdown cells. We found that basal and stimulated AKT1 phosphorylation was not detected in the membrane fraction in PRMT5 knockdown cells, whereas stimulated AKT1 phosphorylation is markedly reduced in GSK591 treated cells (revised Fig. 5a, b). 2) We examined the plasma membrane relocation of AKT1 wild

type or R15K mutant by confocal microscopy, where we found that AKT1 wild type but not the R15K mutant was associated with the plasma membrane (Fig. 5c). 3) We purified HA-tagged AKT1 wild type or R15K mutant proteins and performed *in vitro* lipid binding assay to examine the association of these proteins with phosphatidylserine. We found that AKT1 wild type bound to phosphatidylserine beads, while most of the R15K mutant remained in the flow-through fraction instead of pulldown by phosphatidylserine beads (revised Fig. 5d).

Regarding the actual methylation of AKT, we performed *in vitro* methylation assay where the methylation of AKT1 wild type or R15K mutant was examined by incubation with S-adenosyl-L-[methyl-¹⁴C] with or without PRMT5/MRP50. Our results showed that R15K could not be methylated by PRMT5/MEP50 (revised Fig. 4e). In addition, we pulled down endogenous AKT1 from either DMSO or GSK591 treated cells and analyzed the co-immunoprecipitating proteins using mass spectrometry. However, due to the fact that R15 is close to the N-terminus, which is a common technical challenge in mass spectrometry, this could not be done. Nonetheless, we believe these new data demonstrating the R15K mutation abolishes reactivity with an antibody recognizing SMDA, impairs AKT1 activity and phenocopies the PRMT5 depletion strongly argues this residue is the major methylation site.

3) In figure 3, it is possible that at least some of the reduced AKT phosphorylation is due to effect of the PRMT5 inhibitor on upstream components including RTKs. The heatmap in Fig.3A shows effects on MAPKs and their downstream targets (e.g. p53-Ser15 or pRSK), but no downstream targets of AKT. These point is not addressed in the following figures as well. It is recommended to follow the phosphorylation of known AKT substrates such as GSK3 or Foxo1 in the presence of the PRMT-5 inhibitors to verify that the methylation is indeed required for AKT activity.

Response: We appreciate the reviewer for pointing out this issue. We are sorry if this was not clear in the earlier submission. In the revised submission, the targeted protein array detected the decrease of phospho-GSK3 β Ser9 (Fig. 3a), and this is highlighted in the revised manuscript by adding an arrow in the revised Fig. 3a. In addition, in the revised submission, we have included individual western blots showing that phosphorylation of GSK3 α and GSK3 β was reduced (revised Fig. 3b-e). It has been well documented that PRMT5 regulates cell cycle proteins and thus promoted cell proliferation. Therefore, it is not surprising that phospho-p53 Ser15 levels were reduced upon PRMT5 inhibition. The RTKs on the protein array did not show substantial changes, nor did the known RTKs upstream of AKT (see original Supplementary Fig. 3a/revised Supplementary Fig. 3e). Because Foxo1 levels are very low in neuroblastoma cells, we used GSK3 α and GSK3 β as a readout of AKT activity.

4) The results represent in figure 3 demonstrate the effect of PRMT5 on the basal phosphorylation of AKT, but the effect of stimulated phosphorylation is ignored. Due to the effects on the receptors, it is important to show what is the effect of EGF/FGF/NGF stimulation on the PRMT-5 effects.

Response: We agree with this concern. We have now examined AKT phosphorylation in GSK591 treated cells with or without EGF stimulation and found that PRMT5 is also critical for AKT phosphorylation upon stimulation. These results are added in the revised submission in Fig.

3d.

5) a) In Fig. 4 and Fig S3, the immunoprecipitation (IP) and Co-IP of the AKT isoforms is not demonstrated. The authors should present proper controls for each of the IPs in order to verify that the signals are indeed from the specified isoforms with no cross reactivity between them. This will also verify that the amounts precipitated were equal. Also, a verification of the Ab specificity of each isoform should be done (by siRNA) in a separate experiment. b) As expected, there are changes in the molecular weight of the different AKT isoforms when tested in the CHLA 20 cells. However, no changes are seen in NGP cells, and I wonder if this is reproducible or what might be the reason for it. c) Importantly, it is not clear why the IgG band is not apparent in any of the blots from the Co-IP experiments.

Response: We thank the reviewer for pointing out the critical controls. To address these concerns, in the revised manuscript we have done the following:

Point (a) We purchased these antibodies from Cell Signaling Technology Inc (CST). Upon our request, CST shared their results when they characterized these antibodies (shown in Revision Figure R1). These results show that these antibodies are specific for each AKT isoform.

Point (b) The difference of molecular weight of AKT isoforms is marginal, and the visualization of the bands mainly depends on the protein separation on SDS-PAGE, i.e. the percentage of gel and duration of running time. To further address this concern, we re-did the entire experiment showing that an equal amount of AKT isoforms was pulled down from DMSO and GSK591 treated cells (revised Fig. 4b and Supplementary Fig. 4b, c). These data show SDMA on AKT1 is reduced upon GSK591 treatment.

Point (c) We used a mouse monoclonal anti-rabbit IgG conjugated to horseradish peroxidase (HRP) for western blot, which mainly recognizes IgG light chain instead of the heavy chain. That explains why the IgG heavy chain was not detected in the IP presented in the earlier submission. We show this in an uncropped full blot (Response Fig.3). To further address this concern, we have repeated all IP experiments using a rabbit polyclonal secondary antibody conjugated to HRP for western blot where the IgG heavy chain showed up in each IP (revised Fig. 4b and Supplementary Fig. 4b, c).

6) In continuation to point 5, the amount of precipitated proteins should be shown and quantitated in all IP and CoIP experiments. Input controls are not sufficient in this case.

Response: We agree with the reviewer's comments. We repeated all IP and co-IP experiments and have now included equal amounts of precipitated proteins to each IP experiment in the revised manuscript (Fig.4a, b, Supplementary Fig. 4b, c, Fig. 5e-g).

7) (a)The legend of Fig. 4 is truncated, and therefore the details of the experiments of Fig. 4f are not known. (b)Moreover, the amount of IPed AKT1 is not presented, and the phosphorylation is of AKTs from unstimulated cells. (c) Finally, from what I understand, it is still possible that the effect on AKT1 phosphorylation is not direct, and might be affected by interaction with regulators or affect EGFR expression (as in Fig. 7). This is an essential point that should be nailed by the authors; for example by using an in-vitro phosphorylation with active PDK1, and by examining the role of the mutation on AKT binding to other proteins (interactome). Two other points: (d) it is not clear why the IgG band is not seen in the IP experiment, and (e) why do

the authors use the term AKT1 15G/K, 15R/K, or better, R15K, as appears in page 9. (f) Finally, the reduction of SDMA of AKT-1 by GSK591 (Fig. 4C) does not look significant.

Response: We thank the reviewer for raising these critical questions and apologize for the truncated figure 4 legend.

Point (a) In the revised manuscript, we have fixed this problem.

Point (b) In response to Comment #6 above, we showed equal amounts of AKTs were immunoprecipitated from each sample in the revised manuscript (Fig.4a, b, Supplementary Fig. 4b, c, Fig. 5e-g). In addition, we also showed that both basal and stimulated AKT were compromised under PRMT5 inhibition in revised Fig. 3d.

Point (c) To nail down whether reduced AKT activation is directly caused by PRMT5, or indirectly affected by PRMT5 downstream targets, we showed that only the wild type PRMT5 but not the enzymatic deficient mutant PRMT5 could restore the decreased phosphorylation of AKT (revised Fig. 4b, c). Further, exogenous EGFR failed to rescue this phenotype (revised Supplementary Fig. 3f). In addition, EGFR inhibitor erlotinib treatment did not have a synergistic effect on AKT phosphorylation in GSK591 treated cells (Supplementary Fig. 3g). Moreover, we investigated the known upstream regulators of AKT activation, such as ERBB3, IGF1R, VEGFR, and found that none of them were affected by PRMT5 inhibition (original Supplementary Fig. 3a/revised Supplementary Fig. 3e). Although we cannot completely rule out the possibility that the compromised AKT activation might result from an indirect factor, these results suggest that PRMT5 most likely regulates AKT activation in a direct manner. We believe that further screening of other RTKs that might be affected by PRMT5 inhibition and mediate AKT activation is beyond the scope of this study.

Point (d) Please see our response to Comment #5, above.

Point (e) We agree with the reviewer that the nomenclature should be consistent as AKT1-R15K, and we have made this change in the revised manuscript.

Point (f) We quantified the reduction of SDMA of AKT1 by GSK591 treatment by Image J, which showed SDMA on AKT1 in GSK591 treated cells reduced by more than 80% (Response Fig. 4).

8) (a) In Fig 7, it seems that GSK591 affects all RTKs, indicating that the effect might be global and may affect the degradation or trafficking of the receptors. This point should be better explained. (b) In addition, is there a reasonable explanation as to why the expression of EGFR (in the presence of GSK591) is dramatically reduced in CHLA 20 cells and only moderately in NGP cells? (c) Also, in another experiment (Fig 7E), the effect of GSK591 on EGFR expression in CHLA 20 is also moderate. Which of the two experiments represents the real effect of GSK591? Quantification of the blot should be helpful. (d) Finally, it is not clear to me why MK-2206 affects AKT phosphorylation. It is an AKT inhibitor and is not supposed to inhibit the PDK1 or mTORC2 activity, and indeed we and other colleagues do not detect such a change.

Response: We thank the reviewer for raising these questions.

Point (a) We believe that the reviewer is referring to the results in Fig. 6. As we showed in the original Supplementary Fig. 3a/revised Supplementary Fig. 3e, GSK591 does not affect the protein levels of ERBB3, IGF1R, or VEGFR. In addition, the RNA-seq analysis did not identify significant changes to most RTKs other than EGFR, NGFR, and FGFR4. Moreover, we also showed that exogenous EGFR did not rescue the decrease of AKT phosphorylation in GSK591

treated cells (original Fig. 7b/revised Supplementary Fig. 3f). And the addition of EGFR inhibitor erlotinib failed to further reduce AKT phosphorylation in GSK591 treated cells (original Fig. 7c/revised Supplementary Fig. 3g). All these results argue that PRMT5 most likely directly regulates AKT phosphorylation. As reviewer#2 suggested that the manuscript should have focused on PRMT5/AKT, we have removed PRMT5/EGFR from the revised manuscript to make our manuscript focus on the topic.

Point (b) We also noticed the decrease of EGFR expression in CHLA20 was greater than that of NGP, which might result from the fact that c-MYC is overexpressed in CHLA20 and N-MYC is amplified in NGP. It has been shown that c-MYC binds to the promoter region of EGFR and activates its expression (PMID: 28778566) and c-MYC assists *MALAT1*-KTN1 to positively regulate EGFR protein expression (PMID: 30683916). In RNA-seq, we detected a decrease of c-MYC but not NMYC transcripts upon GSK591 treatment. Therefore, it's possible that EGFR is affected by both direct and indirect mechanisms when PRMT5 is inhibited.

Point (c) We have a short exposure image of the EGFR western blot shown in the original Fig. 7e (presented in Response Fig. 5). This exposure shows a strong effect of GSK591 on EGFR expression, consistent with that shown in original Figure 7e.

Point (d) MK-2206 treatment has previously been shown to decrease AKT phosphorylation in some experimental settings (Mol Cancer Ther. 2010. PMID: 20571069; Leukemia. 2012. PMID: 22614243; Clin Cancer Res. 2012. PMID: 22550167; Cancer Res. 2012. PMID: 22815528; Oncotarget. 2013. PMID: 24036604; Clin Cancer Res. 2014. PMID: 24583795; PLoS ONE 2016. PMID: 27487157; Oncogenesis. 2017. PMID: 28991258; PLoS ONE. 2017. PMID: 22911820; Br J Cancer. 2018. PMID: 30377337; PLoS One. 2018. PMID: 29470540 BMC Cancer. 2019. PMID: 30943918). This may be due to a context-dependent effect. We selected two figures from the literature and have included them in Response Fig. 6.

9) In many cases, the presented data are either not probable or not explained well enough. For example, (a) why don't the authors detect the IgG in the IP experiments, (b) What is the reason for the big difference between the results in Fig. S1a and S1c in the control lanes. (c) The background in all experiments is too homogeneous, which may indicate that the figures were heavily manipulated. Therefore, I suggest to provide the original blots in the supplementary. (d) Finally the molecular weight markers are problematic and not reproducible in many cases. For example, PRMT5 appears in some cases in ~73 kDa, but in other blots (e.g. Fig. 6C) at ~76 kDa. The molecular weight of AKT isoforms should be different between each other. (e) PP2A-C is a 34 kDa protein (not 39 as appear in Fig. S3b), (f) the MW of PHLPP1 and 2 seem to be replaced, g) PP2A-A is 65 kDa and it is not clear what PP2A B is used.

Response: We understand the reviewer's comments and concerns.

Point (a) this issue is addressed in comments #5 point c.

Point (b) Supplementary Fig. 1a had three lanes of control for three different cell lines, CHLA20, SK-N-BE(2), and NGP respectively. Supplemental Fig. 1c showed four lanes of control for non-transduced or transduced with Scramble in NGP and SK-N-BE (2) cells. The different intensity of signal between controls resulted from the amount of lysate loaded in these two experiments by different researchers. All the samples in Supplementary Fig. 1a were run in a 15-well gel, therefore the amount of lysate loaded was less than the one presented in Supplemental Fig. 1c in which all samples were run in a 10-well gel (in which wells have a larger volume than when using 15-well combs).

Point (c) We have uploaded original, uncropped Western blot results as Supplementary Fig. 6 in the revised manuscript to verify the original data. To be absolutely clear: there was no manipulation of these images and all images shown in our figures reflect precisely what is observed in the uncropped scans.

Point (d) We apologize for the confusion of the molecular weight of those proteins shown in the figures. This might come from mistakes during the preparation of figures or the percentage of gel and duration of running time. We have carefully compared the results presented in the figures to the uncropped data and corrected the inaccuracy.

Point (e) The antibody to detect PP2A-C was purchased from Cell Signaling Technology Inc. From the datasheet, the predicted molecular weight of PP2A-C is 36 or 38 KD (Response Fig. 7). Meanwhile, proteins may be bigger than their predicted molecular weight because of context-dependent modification or differences in conditions used for SDS-PAGE.

Point (f) We checked the original films and found the molecular weight of PHLPP1 and PHLPP2 was consistent as shown in original Supplementary Fig. 3b (also see Supplementary Fig. 6-uncropped western blots). We speculate that the confusion may result from the separation of these proteins on SDS-PAGE gels or context-dependent modification of the proteins.

Point (g) Our western blot showed the size of PP2A-A was about in the middle of the range between the 50 kd and 75 kd markers, which is not unreasonable, given its predicted size. As we surveyed the potential involvement of PP2A phosphatase, we checked all three isoforms of PP2A family. We found that the PP2A family phosphatases are not responsible for the reduction of AKT phosphorylation upon PRMT5 inhibition.

10) Several minor points are: (a) in Figure 7D the effect of the phosphorylation of EGFR by WT or mutant AKT-1 is not shown (only the expression is shown). (b) in Figure 4B: Based on the results and in contrast to what is written (page 8, rows: 8-9), the interaction between PRMT5 and AKT (or pAKT) does not change following inhibition of PRMT5. Rather, GSK591 “somehow” interfere with the phosphorylation of AKT (by PDK-1 and mTORC-2). This point should be explained. (c) Do the authors know if PRMT5 plays a role in AKT activation in different cell lines as well? How general is the role of PRMT5 in AKT activation? (d) In the introduction, the explanation of the way AKT is activated is not accurate, and one of the kinases that phosphorylates Ser473 is mTORC2 (not rictor).

Response: We thank the reviewer for these suggestions.

Point (a) GSK591 treatment reduced the transcription of EGFR (original Fig. 6a). Consequently, there is a decrease in EGFR total protein (original Fig. 6b-c). If the total protein of EGFR is down-regulated, the phospho-EGFR should also be reduced. However, it is impossible to discriminate what fraction of the reduction in phosphorylation is due to reduced EGFR levels and what fraction is due to impaired phosphorylation of the EGFR that remains.

Point (b) As the reviewer points out, the interaction between PRMT5 and AKT is not affected by GSK591. From the study of EPZ015666, a sister compound of GSK591, the key cation- π interaction between the tetrahydroisoquinoline (THIQ) benzene ring and the cofactor SAM is thought to contribute to the affinity and selectivity of EPZ015666 against PRMT5. However, EPZ015666 has a competitive advantage with peptide substrate, but not with SAM. Thus, it is likely that GSK5691 prevents PRMT5 from methylating its substrates but does not prevent PRMT5 from binding its substrates. As a result, in the presence of GSK591, PRMT5 will bind

but not methylate AKT, and the failure to methylate AKT will impair its phosphorylation by PDK1 and mTORC2.

Point (c) We have unpublished results in another project that PRMT5 methylates AKT2 in mouse hepatocytes. We believe that PRMT5 plays an important role in regulating AKT activation in general, but the specific arginine substrate is likely to be cell type or tissue type specific. This is an interesting topic to further pursue in our future studies.

Point (d) We appreciate the reviewer pointing this out. We meant to state that Rictor is required for phosphorylation, not that it is the catalytic subunit. We modified the sentence in the revised introduction to more accurately describe mTORC2 function.

Reviewer #2 (Remarks to the Author):

We thank Reviewer #2 for his/her invaluable insight and comments.

This paper identifies a role for PRMT5 in the neuroblastoma (NB) tumour growth and metastatic dissemination and identifies Akt as a novel PRMT5 substrate. In parallel, they find that PRMT5 regulates the expression of growth factor receptors, including EGFR, possibly through epigenetic-mediated regulation. They conclude that PRMT5 promotes NB metastatic growth through independent and Akt and EGFR pathways.

Many of the concepts in this study are not novel. A number of papers have already linked PRMT5 to the phosphorylation and activation of AKT (Zhang et al., J Cell Mol Med, 2019; Chung et al., JBC, 2019; Strobl et al., Mol. Can. Therapeutics, 2020; Li et al., Cell Transplant. 2019)), knockdown of PRMT5 has already been shown to promote apoptosis in SK-N-BE2 cells (Park et al., Molecular Oncology 2015) and, as pointed out by the authors, PRMT5 already has an established role in metastatic tumour cell growth. Yet, the in vivo data demonstrating the potency of GSK591 on NB xenografts growth and liver metastasis is compelling, and the use of these tumours to demonstrate in vivo relevance of in vitro cell culture findings throughout the study is a strong addition.

Unfortunately, I do not find that the overall conclusions regarding mechanistic insight are fully supported by data presented. In line with this, the addition of the rather weak EGFR epigenetic data distracts from the main interesting finding regarding how methylation promotes Akt activation, and feel that if the authors had focused on the latter, the paper would have been more compelling.

Comments:

It appears that administration of GSK595 completely blocks tumour growth, so much so that data presented in Fig 2b and c demonstrate that there is no tumour mass (although there must be something as downstream analysis was conducted). (a) The authors state that GSK595 was administered when tumours reached a certain luciferase pixel intensity, however as this is an arbitrary number it is difficult to know how big tumours were at the start of treatment. (b) I am wondering if the authors observed tumor regression? It would be nice to see data of luciferase intensity at the start of GSK595 treatment and at the end. (c) I am also wondering why the in vivo experiment was not conducted using Sk-N-BE2 cells as this was derived from a relapsed tumour. If GSK595 has an effect in this cell line, this could have significant clinical implications for second line therapy options. (d) What happens to tumour growth if GSK595 is administered after

tumours have grown to a well-established size?

Response: We thank the reviewer for his or her thoughtful questions regarding the in vivo studies.

Point (a) The xenograft model we used is to inject a million cells in the kidney renal capsule, the site where neuroblastoma was originally derived. Because we can't measure the kidney with the tumor when animals are alive, therefore, we have to use live animal imaging to confirm the tumor establishment. Also, due to challenges with this technique, tumor growth is not as homogenous as seen in other xenograft models such as subcutaneous injection. Therefore, we randomized the animals with similar bioluminescent signals to each group. The tumor mass from GSK595 treated group is not zero (<0.1 gram) but is significantly smaller than from the vehicle control group. In these experiments, luciferase intensity started at $\sim 10^7$ pixel per second (p/s) and increased to $>10^9$ - 10^{10} p/s in control mice by the end.

Point (b) We have included in the revised Table-3 the data of bioluminescent intensity prior to the treatment and at the end of the experiments. In the xenograft studies presented in Fig. 2, we did not observe tumor regression upon GSK595 treatment, but a significant reduction of tumor growth and metastasis.

Point (c) We chose CHLA20 and NGP to represent MYCN non-amplified and MYCN amplified neuroblastoma. It is worth noting that CHLA20 was derived from a relapsed tumor. Moreover, we conducted new animal studies in a metastasis mouse model presented in the revised manuscript, where inducible PRMT5 knockdown SK-N-BE(2) cells were injected by tail vein (Fig. 7, Supplementary Fig. 5a-d). We found that PRMT5 depletion in SK-N-BE(2) blocks tumor metastasis in the liver and lung.

Point (d) The kidney renal capsule implantation model faithfully recapitulates the aggressiveness of human neuroblastoma where the tumor grows quickly after it has been established. For the data presented in Figure 2, we started the treatment after the tumors were established after bioluminescent intensity increasing from 10^5 p/s (a week after implantation) to 10^7 p/s (15 days and 9 days post-implantation for CHLA20 and NGP respectively). The time window is about two weeks before the tumor burden reaches the maximal IACUC allowance. Therefore, it may not be feasible to perform the experiment in this manner.

I find the colour scale of Fig 3a rather confusing. I am assuming that green indicates a decrease in protein expression/PTM, however the scale goes from 0-9. Most of the targets are a white bar, which I am assuming that this means no change after DMSO treatment. Shouldn't this be zero?

Response: We thank the reviewer for pointing out this confusion. To better illustrate the results, we re-analyzed the data and now present it as log₂ fold change over DMSO. The green or red color indicates the decrease or increase respectively, whereas the white color means no change (revised Fig. 3a).

At the start of the "PRMT5 methylates AKT" section, the authors list a number of proteins involved in AKT signaling. (a) Are any of these genes mutated in the cell lines used? (b) Are the basal levels of AKT signaling increased compared to non-transformed cells? (c) Following on from this, is PRMT5 required for growth factor mediated AKT signaling? This will enable a distinction between PRMT5 regulating basal or induced AKT signaling. (d) Likewise, it may be that proteins blotted for in Supp Fig 3 are PRMT5 dependent after stimulation.

Response: We agree that these are important considerations.

Point (a). We looked at the next-generation sequencing data of NGP and SK-N-BE (2) in the Broad Institute DepMap Portal (<https://depmap.org/portal/>) and found that there is no mutation in these genes. However, we could not find whole-genome mutation analysis in the CHLA20 cell line in the public databases. Therefore, we did a bioinformatic analysis of the TARGET database and did not find reports of mutations in these genes in neuroblastoma patients. It's most likely that these genes are not mutated in the cell lines used in this study.

Point (b) Neuroblastoma arises from neural crest cells that are transient during development. Therefore, it does not have a matched non-transformed tissue for the comparison of whether AKT signaling is dysregulated. In general, changes in the expression of AKT are not considered a major regulatory mechanism of these proteins, which might be expressed in a constitutive manner. Contrarily, posttranslational modifications (and subsequent localization) are the most extensively studied and well-accepted mechanism of functional modulation. In addition, most AKT downstream targets are regulated on posttranslational levels, such as GSK3 α and GSK3 β . When we analyzed the transcripts of AKTs, GSK3 α , and GSK3 β in high-risk versus low-risk neuroblastoma patients in the R2 neuroblastoma patient database, we did not see that these genes are differentially expressed on transcriptional levels.

Point (c) We have now included new data in the revised Figure 3d showing that PRMT5 is also required for EGF stimulated AKT activity.

Point (d) We examined the expression of PDK1, Rictor, PTEN, and phosphorylation of PDK1 in GSK591 treated cells with or without EGF stimulation and found no change in these targets. We have included these new data in revised Supplementary Fig. 3d.

The finding that GSK591 suppresses Thr308 and 473 phosphorylation should be substantiated by knockdown analysis.

Response: These results were presented in the original Figure 3c.

(a) Why does GSK591 not reduce the apparent SDMA on AKT2/3? (b) How convinced are the authors on the specificity of their SDMA antibody (I could not find details of it the antibody table). (c) Are AKT2/3 methylated by PRMT9? (d) Experiments should to be repeated using shPRMT5 and in vitro methylation assays with recombinant proteins.

Response:

Point (a). This is an interesting question. Although AKT2 and AKT3 are highly expressed in neuroblastoma cell lines used in this study, the methylation of these proteins is not affected by PRMT5 inhibition. We have unpublished data in another project that PRMT5 methylates AKT2 but not AKT1 in mouse liver cells. This may suggest that the selection of PRMT5 substrates is cell type or tissue type specific. We now added text to the Discussion on page 9 lines 18-26 to address this point.

Point (b) Cell Signaling Inc. characterized their SDMA antibody and generously shared the results with us. These results are added in Response Figure R2. Moreover, we also tested another SDMA antibody from Millipore (SYM10), and it showed similar results. We have included this result in revised Supplementary Fig. 1d.

Point (c) There are two genes termed PRMT9, FBXO11/PRMT9 (Gene ID: 80204), and PRMT9 (Gene ID: 90826). Mark Bedford's group characterized the latter with solid evidence that it is a type II arginine methyltransferase (PMID: 25737013). We depleted PRMT9 expression by siRNA in CHLA20 cells and found that it did not affect AKT phosphorylation (revised Supplementary Fig. 4d). We discuss this point on page 9 lines 18-23 in the revised manuscript. Point (d) We performed an *in vitro* methylation assay using the well-established protocol from Dr. Mark Bedford. We transfected AKT1 wild type or R15K mutant in CHLA20 cells and pulled down the exogenous proteins by HA beads. Then the precipitated proteins were incubated with ¹⁴C labeled Adenosyl-L-Methionine (SAM) with or without recombinant RMT5/MEP50. We found that only the AKT1 wild type but not the R15K mutant was methylated by PRMT5/MEP50, and no signal was detected in the sample where AKT1 wild type alone was incubated with ¹⁴C-SAM, indicating there are no other proteins mediating this methylation event. We have included this new data in revised Fig. 4e.

Data in Figure 4b and accompanying supplementary is not convincing. The authors should use PLA for co-localisation analysis.

Response: We agree with the reviewer that those results did not demonstrate the co-localization of PRMT5 with AKT. We, therefore, performed the immunofluorescence in neuroblastoma cells transfected with AKT1 wild type or R15K mutant with or without EGF stimulation by confocal microscopy. We found that AKT1-R15K mutant failed to translocate to plasma membrane even under EGF stimulation. We have included the new results in the revised manuscript Fig. 5c. These data further support that PRMT5 inhibition attenuates basal as well as stimulated AKT phosphorylation.

In many figures, after immunoprecipitation the western blot of protein used as bait is omitted (e.g. Fig 4a). This must be included.

Response: We repeated each IP experiment with bait protein and have now included in the revised manuscript Fig. 4a,b and Supplementary Fig. 4b, c. (please see above response to Reviewer #1 questions 5 and 6). To further address the question in Fig. 4a, we have now included not only the bait proteins (PRMT5, AKT1) but also GSK3 β and BRG1 as positive controls for AKT or PRMT5 IP respectively.

If PRMT5 is required for Thr308/473 phosphorylation, one would expect reduced phosphorylation of downstream substrates. Why was this not shown?

Response: We examined the expression of AKT downstream targets GSK3 α and GSK3 β upon PRMT5 inhibition and have included these results in the revised Fig. 3b-e. Indeed, we do observe reduced phosphorylation of these substrates.

A major criticism of this paper is that the mechanism by which AKT methylation leads to phosphorylation has not been demonstrated. Does methylation recruit activating kinase?

Response: We agree with the reviewer's comment that the mechanism by which AKT methylation

leads to phosphorylation is a critical point. AKT activation involves its recruitment to the plasma membrane by PIP3 through direct interaction with the PH domain and phosphorylation by kinases such as PDK1 and mTORC2. We performed the following experiments to investigate the mechanism in the revised manuscript. (a) examined the levels of phospho-AKT1 in the cytosolic and membrane fraction extracted from DMSO or GSK591 treated cells by Western blotting (Fig. 5a); (b) measured phospho-AKT1 by immunoblotting in cytosolic and membrane fraction isolated from control or PRMT5 knockdown cells Fig. 5b); (c) investigated the co-localization of AKT1 wild type or R15K mutant with plasma membrane by confocal fluorescent microscopy (Fig. 5c); (d) determined the association of AKT1 wild type or R15K mutant with phosphatidylserine by lipid/protein pulldown assay (Fig. 5d); (e) examined the interaction between AKT1 wild type or R15K mutant with kinases PDK1 and mTORC2 (Fig. 5e), or endogenous AKT1 with PDK1 and mTORC2 in PRMT5 knockdown cells (Fig. 5f), and in DMSO or GSK591 treated cells, respectively (Fig. 5g). In summary, PRMT5-mediated AKT1 Arg 15 methylation is required for AKT1 association with the plasma membrane, more specifically binding to phosphatidylserine, and recruitment of PDK1 and mTORC2.

The second major criticism is the inclusion of the RNA-seq and EGFR promoter analysis: it feels like the authors had two separate stories that they have tried to bring together. More importantly, data presenting is not convincing. Why were two different cell lines use for 6f and g? Why was IgG control ChIP not included? Why was H3R3me2s not examined, particularly as this is the recognised PRMT5 epigenetic mark that promotes gene expression, not H4R3me2s, which generally correlates with gene repression? Why was EGFR activation not specially looked at rather than the unconvincing data in Fig 6e?

Response: We agree with the reviewer that adding the EGFR data to the main story of AKT methylation seems redundant, especially after we generated a substantial body of new results on the mechanism of R15 methylation on AKT phosphorylation. Further, without a global PRMT5 ChIP-seq, the existing data presented in original Figure 6 are premature to draw conclusions about the mechanism by which PRMT5 regulates EGFR signaling. Therefore, we removed the EGFR story from the revised manuscript.

Fig 7A - I do not understand the rationale of ectopic PRMT5 expression in GSK591 expressing cells. Surely it makes more sense to do the experiment in shPRMT5 cells?

Response: We agree with the reviewer that a rescue experiment in shPRMT5 cells is better. We performed these rescue experiments in PRMT5 knockdown cells, with similar results (revised Fig. 4c).

Figure legends are difficult to follow. Most of the times the signpost letter is placed at the end of the sentence rather than the front, and then it flips.

Response: We are sorry for the confusion and have now re-edited the figure legends.

Minor points:

Can more information be provided in the text regarding the proteomics-based target screen in Fig 3?

Response: This was done by Activsignal (<https://www.activsignal.com/>). There are not many details released from the company on its public platform presumably due to the concern of intellectual property. By private communication, we were informed that this assay is a similar approach to PLA assay but with modification and in a high throughput manner.

Gene identified in the overlap of the two cell lines should be listed (Fig 5b).

Response: We have included a list of the top 20 overlapping genes in Response Figure 8. Since we have taken out the EGFR story, this is not included in the revised manuscript.

Catalogue number for H3R8me2s and H4R3me2s repeated twice in table.

Response: We thank the reviewer for pointing out this error. The table of antibodies was updated in line with the revised manuscript.

Not sure what the “dimer EGFR” and arrow to “cancer Cells” in the schematic signifies in relationship to this study.

Response: The graphic summary was updated based on the revised manuscript.

Figure ledged 3: typo – full cell name for BE2 omitted.

Response: This typo was fixed in the revised manuscript.

Reviewer #3 (Remarks to the Author):

These authors show that PRMT5 promotes neuroblastoma metastasis by increasing the transcription of EGFR and elevating AKT1 activation by methylation on Arg 15. I thought the biochemistry was sound, however the mouse in vivo studies could be improved.

(a) Specifically, can survival curves be added for Fig 2. (b) Can this compound be analyzed in PDX models in addition to the single cell line xenograft shown? (c) The data that PRMT5 is a driver of metastases is not particularly strong. Can metastases be examined also in PDX studies in vivo? (d) Do levels of PRMT5 correlate with metastases specifically in human tumor databases, and doe introduction of PRMT5 into cell lines and PDXs drive metastases?

Response: We thank the reviewer for these thoughtful questions regarding animal studies.

Point (a) The experiments presented in Figure2 were designed as an endpoint approach to objectively measure and determine whether the treatment was beneficial. While a survival curve is a complementary approach that would certainly improve the value of the data, we were unable to obtain the data in the original experimental settings. To this end, we could not add the survival cures to figure 2.

Point (b) We showed in the original manuscript two animal studies using two different cell lines representing MYCN non-amplified and amplified types of neuroblastomata. While PDX models recapitulate the biological and molecular characteristics and heterogeneity of the patient's tumor and have been implicated in directing personalized treatment, emerging evidence has also suggested limitations of PDX models in cancer research, such as limited engraftment rate, variable phenotypes, genomic instability, and molecular inconsistency (Cancer Res. 2015. PMID: 26180079; Cells. 2019. PMID: 31226846; Int J Cancer. 2020. PMID: 31479514). Furthermore, implantation of patient-derived tumors in immunodeficient mice as we did with cell line xenografts would still leave a big gap of knowledge regarding the potential role of the PRMT5/AKT1 axis in the tumor microenvironment. As our manuscript aims at providing an insightful picture of PRMT5 in neuroblastoma, we feel that PDX models might add relatively modest insights at a prohibitive cost/time.

Point (c) We agree with the reviewer's comment. Therefore, we performed an additional animal study to strengthen the link between PRMT5 and metastasis. We used inducible shPRMT5 systems to knock down PRMT5 in SK-N-BE(2) cells in a mouse metastasis model by tail vein injection. We analyzed the tumor cell in the liver, kidney, lung, and bone marrow as a readout for metastasis. These data are presented in new Figure 7 and Supplementary Fig. 5a-d. These new data provide stronger evidence that PRMT5 depletion impairs metastasis.

Point (d) We could not find data showing PRMT5 expression in metastatic neuroblastoma within any public database. We did an *in vitro* trans-well invasion assay where PRMT5 wild type or enzymatic activity deficient mutant was introduced to CHLA20 cells. The results showed that overexpressing PRMT5 wild type but not the mutant increased the expression of SNAIL and TWIST1 as well as cell invasion through ECM. We have included these results in revised Fig. 6i, j.

2. In western blots in Fig 3, were cells examined at steady state? Was serum or were growth factors added prior to lysis? Supplementary Fig 3, although a negative result, is a very convincing and clean figure.

Response: We thank the reviewer for the recognition of our efforts on those negative results. For the western blots shown in original Figure 3, cells were examined at a steady state. We added a new set of data in revised Fig. 3d showing the changes of AKT activation in DMSO or GSK591 treated cells serum-starved overnight with or without EGF stimulation.

3. I am not sure I understand the rescue experiments in Fig 7. If you are treating cells with inhibitor, then why doesn't inhibitor block the activity of the transfected PRMT5 in addition to the endogenous protein?

Response: We agree with the reviewer that the presence of GSK591 would inhibit both endogenous as well as exogenous PRMT5. However, the forced expression of exogenous PRMT5 driven by CMV promoter outweighed the effect of inhibitor. To avoid the interference of GSK591 on exogenous PRMT5, we performed the rescue experiments in PRMT5 knockdown cells (revised Fig. 4c). Re-expressing PRMT5 wild type but not the enzymatic activity deficient

mutant completely rescued the decreased AKT phosphorylation. This new data is included in revised Fig. 4c.

Fig. R1: Characterization of AKT isoform-specific antibodies

Fig. 1a: Western blot analysis of extracts from Akt1, Akt2 and Akt3 knock-out MEFs and matched wild-type MEFs using Akt1 (#2938) (upper) and Akt2 Antibody #2962 (lower). Lane 1, Akt1 knockout MEF; Lane 2, Akt2 knockdown MEF; Lane 3- Akt 3 knockout MEF; Lane 4- wild type MEF.

Fig. 1b: Western blot analysis of recombinant Akt1, Akt2 and Akt3 proteins using Akt1 (#2938) (upper) and Akt (#4685) (lower).

195 citations, including several from 2020

Fig. 1c: Western blot analysis of extracts from various cell lines using Akt2 Rabbit mAb (#3063) (upper) and Akt 1 #2938 (#2938) (lower).
93 citations on the product page, 98 in CiteAb

Fig. 1d: Western blot analysis of extracts from various cell lines using Akt3 Mouse mAb (#8018) (upper) and Akt (pan) Mouse mAb (#2920) (lower).

Fig. 1e: Western blot analysis of recombinant Akt1, Akt2 and Akt3 proteins show isoform specificity Akt3 Mouse mAb (#8018) (left) and Akt (pan) Rabbit mAb (#4691) (right).

10 citations as recently as 2019 in various peer-reviewed journals

Fig. R2: Characterization of symmetric dimethyl-arginine antibody mix

Fig. 2a: Western blot analysis of MCF7 cells, untreated (-) or treated with Adenosine-2', 3'-dialdehyde (AdOx, 100 μM, 24 hr; +) using Symmetric Di-Methyl Arginine Motif [sdme-RG] MultiMab™ Rabbit mAb (#13222) mix (upper) and GAPDH (D16H11) XP® Rabbit mAb #5174 (lower).

Fig. 2b: The specificity of Symmetric Di-Methyl Arginine Motif [sdme-R] MultiMab™ Rabbit mAb mix was determined by peptide ELISA. The figure demonstrates that the antibody is specific for Symmetric Di-Methyl Arginine and does not react with mono-methyl, di-methyl or tri-methyl lysine and does not react with mono-methyl or asymmetric di-methyl arginine.

Fig. R3: Characterization of symmetric dimethyl-arginine antibody mix

IP: anti-AKT1 antibody CST (#2938)
 WB with primary antibody: anti-AKT1 antibody CST #2938
 Secondary antibody: rabbit polyclonal anti-Rabbit IgG, HRP-linked antibody, Cell Signaling Technology CST #7.74

IP: anti-AKT1 antibody CST (#2938)
 WB with primary antibody: anti-AKT1 antibody CST #2938
 Secondary antibody: mouse monoclonal anti-Rabbit IgG-HRP conjugated antibody, Santa Cruz Biotechnology sc-2357

Fig. R4: Quantification of AKT1 SDMA

Fig. R5: EGFR expression in CHLA20

Fig. R6: MK-2206 on AKT Phosphorylation

MK-2206, an allosteric Akt inhibitor, enhances antitumor efficacy by standard chemotherapeutic agents or molecular targeted drugs in vitro and in vivo.

Hirai H, Sootome H, Nakatsuru Y, Miyama K, Taguchi S, Tsujioka K, Ueno Y, Hatch H, Majumder PK, Pan BS, Kotani H.

Mol Cancer Ther. 2010 Jul;9(7):1956-67. doi: 10.1158/1535-7163.MCT-09-1012.

Epub 2010 Jun 22.

MK-2206, a novel allosteric inhibitor of Akt, synergizes with gefitinib against malignant glioma via modulating both autophagy and apoptosis.

Cheng Y, Zhang Y, Zhang L, Ren X, Huber-Keener KJ, Liu X, Zhou L, Liao J, Keihack H, Yan L, Rubin E, Yang JM.

Mol Cancer Ther. 2012 Jan;11(1):154-64. doi: 10.1158/1535-7163.MCT-11-0606. Epub 2011 Nov 4.

Fig. R7: Molecular Weight of PP2A-C

PP2A C Subunit Antibody #2038

REACTIVITY H M R Mk Dm

SENSITIVITY Endogenous

MW (kDa) 36, 38

SOURCE Rabbit

Fig. R8: Overlapping gene list in RNA-seq

Gene	CHLA-20			NGP	
	log2FoldChange	padj		log2FoldChange	padj
CCND1	0.52	2.6E-74		0.68	1.5E-76
CDKN1A	0.69	1.0E-43		0.68	9.8E-58
MIR34AHG	0.85	2.1E-03		1.58	1.7E-32
LINC00641	0.83	5.3E-48		1.13	9.5E-30
FUS	0.86	2.5E-86		0.50	4.5E-27
UTP14A	1.08	7.6E-31		0.78	4.4E-23
CERK	0.80	4.6E-47		0.51	3.0E-19
TFRC	0.93	4.4E-88		0.53	1.2E-18
DGCR8	0.50	9.3E-07		0.78	8.6E-14
CNTN2	0.58	1.5E-15		0.67	2.0E-13
JUN	-1.07	2.1E-69		-1.18	8.7E-172
ID1	-0.79	2.5E-28		-1.05	8.5E-101
FOS	-2.31	1.1E-44		-1.15	2.1E-72
SNAI1	-0.67	4.6E-16		-1.08	7.4E-69
H1F0	-1.00	2.2E-140		-0.85	1.1E-64
ID2	-0.61	7.3E-22		-0.63	3.5E-63
DDIT3	-1.49	3.1E-23		-1.46	1.7E-45
ACSS2	-0.59	1.6E-08		-1.18	7.0E-29
MTA1	-0.72	4.6E-64		-0.52	2.3E-28
MED15	-0.61	9.1E-18		-0.61	9.5E-27

REVIEWERS' COMMENTS

Reviewer #1 (Remarks to the Author):

The authors successfully addressed my concerns. I have no further comments

Reviewer #2 (Remarks to the Author):

This is a much improved manuscript, and I very much appreciate the effort the authors have made in addressing the comments and the substantial additional experiments that have now been included. I am also happy that they decide to remove the epigenetic EGFR section and focus on understanding how R15 methylation regulates AKT1 activity. This has now provided interesting new insight and the manuscript reads very well. The extra data and discussion surrounding AKT1-R391 methylation is a welcome addition.

Minor points:

Can the colour scheme on Fig 3a be changed so not to disadvantage red/green colour-blind readers.

P6, line 28. Please remove the word “significantly” as no stats test has been conducted.

Figure 5d is missing. This is a shame as this is a very important piece of data that supports the mechanistic insights provided. Can the editor please check this data?

P8, line 34-38: I agree that R15 methylation is required for basal AKT1 activity, but after EGF stimulation, there is a small amount of AKT1 methylation in the R15K expressing cells that translates to a slight induction of AKT1 phosphorylation. It therefore appears that R15 methylation is required for basal activity, is the dominant site for activation after EGF, but is not absolutely essential (at least when detecting activity through immunoblotting approaches). The discussion in this section is therefore too strong, this single PTM is not “essential” for AKT1 activation. Although out of scope of this manuscript, it would be interesting to see what happens with a R15/R391K mutant.

P9, line 16 typo – supp 5e (not 5f).

Can a more extensive figure legend be provided for Fig.8?

Reviewer #3 (Remarks to the Author):

see below

(Note to Editor redacted.)

Response to reviewers' comments

We appreciate the time and efforts spent by the editor on our manuscript and we thank all the three reviewers for their careful review, critical comments, and insightful suggestions.

REVIEWERS' COMMENTS

Reviewer #1 (Remarks to the Author):

The authors successfully addressed my concerns. I have no further comments

Response: Thank you for comments on our work.

Reviewer #2 (Remarks to the Author):

This is a much improved manuscript, and I very much appreciate the effort the authors have made in addressing the comments and the substantial additional experiments that have now been included. I am also happy that they decide to remove the epigenetic EGFR section and focus on understanding how R15 methylation regulates AKT1 activity. This has now provided interesting new insight and the manuscript reads very well. The extra data and discussion surrounding AKT1-R391 methylation is a welcome addition.

Minor points:

Can the colour scheme on Fig 3a be changed so not to disadvantage red/green colour-blind readers?

Response: We thank the reviewer for the recognition of our efforts. We changed the colour scheme in Fig 1f, 1j, Fig 2b, 2e, 2j, Fig 3a, Fig 6e, 6g, 6j, and Fig 7d, 7e.

P6, line 28 Please remove the word “significantly” as no stats test has been conducted.

Response: We removed the word “significantly” from the submitted manuscript.

Figure 5d is missing. This is a shame as this is a very important piece of data that supports the mechanistic insights provided. Can the editor please check this data?

Response: Figure 5d was shown in the previously revised manuscript.

P8, line 34-38: I agree that R15 methylation is required for basal AKT1 activity, but after EGF stimulation, there is a small amount of AKT1 methylation in the R15K expressing cells that translates to a slight induction of AKT1 phosphorylation. It therefore appears that R15 methylation is required for basal activity, is the dominant site for activation after EGF, but is not absolutely essential (at least when detecting activity through immunoblotting approaches). The discussion in this section is therefore too strong, this single PTM is not “essential” for AKT1 activation. Although out of scope of this manuscript, it would be interesting to see what happens with a R15/R391K mutant.

Response: We thank Reviewer#2 for the insightful comments and agree with his suggestion. We toned down the importance of R15 methylation in the discussion section. Because AKT1 R391K mutant is not stably expressed in neuroblastoma cells even in the presence of MG132, we think that R391

probably is constitutively methylated. While the methylation of R15 in the PH domain is primarily responsible for AKT1 association with the plasma membrane, additional methylation on arginine sites in the kinase domain (such as R391) may fine-tune the kinase activity of AKT1. Thus, we speculate that the R15/R391K double mutant may be further impaired for its activation and kinase activity.

P9, line 16 typo – supp 5e (not 5f)

Response: We thank the reviewer for pointing out this mistake. We fixed this issue in the revised manuscript.

Can a more extensive figure legend be provided for Fig.8?

Response: We extended the legend of Fig 8 in the revised manuscript.

Reviewer #3 (Remarks to the Author):
see below

Response: Thank you for the comments on our work.